ecology, evolution, physiology

climate change, partial migration, brown trout, acclimation, *Salmo trutta*, metabolism

**Author for correspondence:**
Louise C. Archer
e-mail: l.archer@umail.ucc.ie

# Associations between metabolic traits and growth rate in brown trout (*Salmo trutta*) depend on thermal regime

Louise C. Archer[1,2,†], Stephen A. Hutton[1,2], Luke Harman[1,2], W. Russell Poole[3], Patrick Gargan[4], Philip McGinnity[1,3] and Thomas E. Reed[1,2]

[1]School of Biological, Earth and Environmental Sciences, University College Cork, Distillery Fields, North Mall, Cork, Ireland
[2]Environmental Research Institute, University College Cork, Lee Road, Cork, Ireland
[3]Marine Institute, Furnace, Newport, Co. Mayo, Ireland
[4]Inland Fisheries Ireland, 3044 Lake Drive, Citywest Business Campus, Dublin D24 Y265, Ireland

(iD) LCA, 0000-0002-1983-3825

Metabolism defines the energetic cost of life, yet we still know relatively little about why intraspecific variation in metabolic rate arises and persists. Spatio-temporal variation in selection potentially maintains differences, but relationships between metabolic traits (standard metabolic rate (SMR), maximum metabolic rate (MMR), and aerobic scope) and fitness across contexts are unresolved. We show that associations between SMR, MMR, and growth rate (a key fitness-related trait) vary depending on the thermal regime (a potential selective agent) in offspring of wild-sampled brown trout from two populations reared for approximately 15 months in either a cool or warm (+1.8°C) regime. SMR was positively related to growth in the cool, but negatively related in the warm regime. The opposite patterns were found for MMR and growth associations (positive in warm, negative in the cool regime). Mean SMR, but not MMR, was lower in warm regimes within both populations (i.e. basal metabolic costs were reduced at higher temperatures), consistent with an adaptive acclimation response that optimizes growth. Metabolic phenotypes thus exhibited a thermally sensitive metabolic 'floor' and a less flexible metabolic 'ceiling'. Our findings suggest a role for growth-related fluctuating selection in shaping patterns of metabolic variation that is likely important in adapting to climate change.

## 1. Introduction

As the fundamental biological rate determining resource use and energy balance [1], metabolism underlies organism performance, life histories, and ultimately, fitness [2]. Metabolic traits—standard metabolic rate (SMR), maximum metabolic rate (MMR), and aerobic scope (AS)—can vary dramatically within species, but for reasons that remain obscure [3]. The baseline energetic demands of ectotherms are defined by SMR, which represents the minimum energetic costs of maintaining tissues and homeostasis in an organism that is inactive, unstressed, and non-digestive [4] (termed basal metabolic rate (BMR) in endotherms within their thermoneutral zone, i.e. requiring minimal changes in metabolic heat loss/gain). MMR in contrast, refers to the highest rate of aerobic metabolism (i.e. oxygen transport and ATP production) that can be achieved [5]. AS—the difference between an organism's SMR and MMR—determines the potential energy that can be allocated towards important functions including digestion, activity, growth, and reproduction [6,7]. Uncovering sources of variation in these metabolic traits and understanding the links between metabolism, performance, and environmental conditions is widely recognized as being crucial to forecasting biological responses to global change [8].

†Present address: Laboratory of Quantitative Global Change Ecology, Department of Biological Sciences, University of Toronto Scarborough, Toronto, Ontario, Canada.

SMR, the most frequently measured metabolic trait, shows substantial inter-individual variation, with up to threefold differences in SMR even among similarly sized and aged individuals from the same population [2]. Although MMR (and consequently AS) is measured less often, the level of variation appears similar to that of SMR, once age and size are accounted for [9]. While SMR and MMR are often correlated within species [10], the relationship between metabolic traits can also vary considerably [11,12].

From an ultimate perspective, temporal and/or spatial variations in factors such as temperature, food, and habitat quality have been proposed to maintain intraspecific variation in metabolic traits via fluctuating selection [13–16]. Genetic decoupling of metabolic traits can also occur across time or space if each trait is subject to distinct selection pressures [5,11]. From a proximate perspective, metabolic traits also exhibit phenotypic plasticity with respect to various environmental factors such as temperature [17], hypoxia [12], food availability [18,19], and habitat structure [20]. Since aerobic metabolism is dependent on factors influencing oxygen demand and uptake, temperature profoundly determines metabolic rates [1,3,14]. In ectotherms, temperature effects on metabolism can be immediate (acute) or can emerge over more prolonged periods (chronic) via long-lasting acclimation responses to changes in thermal regime [8]. Acute effects are reasonably well researched [14,21], but we know less about the effects of chronic temperature exposure on different metabolic traits despite acclimation responses being highly relevant in the context of climate change [7].

Predicting the effects of long-term temperature is made more complicated by the various components of metabolism showing different sensitivities to environmental conditions. There is some evidence for more plastic metabolic 'floors' (SMR) than 'ceilings' (MMR) in response to temperature, as documented in European perch *Perca fluviatilis* that reduced SMR after long-term warming with no compensatory adjustments seen in MMR [22], with a similar response seen in three-spine sticklebacks *Gasterosteus aculeatus* [15]. Disparities in the relative responses of BMR and MMR (or cold-induced maximum metabolism) to temperature are also apparent in endotherms [23–25], indicating that plastic decoupling of metabolic traits may be widespread. While a positive relationship between SMR and MMR has been proposed under the 'increased intake' hypothesis—where a higher SMR maintains the metabolic machinery that fuels high MMR [2,26]—covariation between metabolic components may be stronger, weaker, or non-existent depending on the environmental context [12,27], and relationships can differ among and within individuals [28]. It seems that SMR and MMR might thus be subject to subtly different proximate or ultimate constraints that might be revealed or masked by a given set of environmental conditions [5,27].

Previous studies on the effects of metabolic traits on various fitness metrics have provided mixed results [2,29,30]. Positive relationships between SMR and growth [31,32], reproduction [33], and survival [34] imply fitness benefits of higher SMR that are in line with the 'increased intake' hypothesis. Yet SMR/BMR has also been negatively linked to growth [28,29,35], reproduction [36], and survival [35,37,38], supporting an alternative 'compensation' hypothesis, whereby a lower SMR is advantageous for energy-saving purposes [2]. These inconsistencies are likely explained by context-dependent fitness benefits of metabolic traits. For example, higher SMR may be beneficial when resources are plentiful [19,39,40] or predictable [41], but have negative [42], or no effect [35] when resources are limited. While MMR–fitness associations are reported less often, the few studies that have looked at this suggest that MMR also shows inconsistent relationships with fitness [43–45]. Research has focused largely on the role of resource availability as a selective agent, but other abiotic or biotic factors might also influence relationships between metabolic phenotypes and fitness components. Given the importance of temperature for the energetic process, thermal regime is potentially an important mediator of metabolic rate–fitness links, yet this has rarely been tested [30] despite the imperative of widespread climate warming.

Here, we experimentally reared F1 offspring from two wild trout populations under different thermal regimes to explore how metabolic traits and their relationships with growth (a key fitness-associated trait) are mediated by chronic temperature increases. Specifically, we aimed to (i) test whether long-term temperature elevation leads to variation in SMR and MMR, (ii) explore how metabolic traits are related to growth under different thermal regimes, and (iii) test whether the covariation of metabolic traits is influenced by the thermal regime. We expected that chronic temperature elevation would lead to compensatory responses in metabolism (lower SMR or higher MMR). We also expected that relationships between metabolism and growth might depend on the thermal regime, whereby SMR and MMR would generally show a positive relationship to growth (under a natural thermal regime) as per the increased intake hypothesis [2], but that long-term temperature elevation might result in a negative SMR–growth relationship (if relatively high SMR are disadvantageous in warmer environments because less energy remains for growth once maintenance costs are paid, in line with the compensation hypothesis [2]). We expected that the MMR–growth relationship could either stay positive in the warmer regime, or alternatively could become neutral or negative (if SMR and MMR are tightly coupled, and lower SMR also results in reduced MMR).

## 2. Methods

### (a) Study populations and fish rearing

In November 2015, we obtained brown trout brood stock from two wild populations in the west of Ireland by seine netting in the Tawnyard Lough (56 ha) in the Erriff catchment (53°37′ 0.00″ N: 09°40′ 17.10″ W) and in the Srahrevagh river in the Burrishoole catchment (53°57′ N: 09°35′ W) (electronic supplementary material, figure S1). Four males and three females were used as brood stock from Tawnyard, and 12 males and females from Srahrevagh. The populations vary in life-history tactics, with anadromy (sea-migration) frequent in the Tawnyard population [46] (termed the 'anadromous-background population') and relatively rare in the Srahrevagh population [47] (the 'non-anadromous-background population').

See [48,49] for a detailed description of crossing, fertilization, and rearing procedures. In brief, eggs from each female were fertilized by 1–2 males from the same source population. Post-hatching, fry were reared in 100 l growth tanks (one per population) on a recirculating aquaculture system (RAS) at University College Cork (Aquaculture and Fisheries Development Centre), Ireland. Fish were fed ad libitum with commercially available trout pellets (Skretting Ltd., Norway) and were maintained at a natural temperature regime and constant photoperiod (12 : 12 h of light : dark), until experimental treatments began.

## (b) Temperature treatments

In December 2016, fry were allocated to four 203 l capacity tanks in a larger experimental RAS ($n = 35$ per tank) with the populations reared separately throughout the study (i.e. each population allocated across two of the four tanks). LED lights above each tank simulated the natural photoperiod of the source catchments. Fish were fed daily pellet rations for optimal growth calculated as a percentage of body mass as per manufacturer's instructions, with absolute rations adjusted monthly to account for changing temperatures and body mass. Automatic feeders above each tank delivered daily feed in multiple localized pulses. Excess feed removed during cleaning indicated fish were feeding to satiation. Water quality was consistently within acceptable levels for fish health, with great care taken to ensure that all measured variables other than temperature regime (fish densities, feeding, photoperiod, lux, and flow rates) were constant across tanks. Mortality was negligible, but fish were haphazardly culled ($n = 20$) over the course of tank rearing for inclusion in parallel studies.

Each of the four tanks was allocated to one of two temperature treatments in January 2017, with one warm and one cool tank for each population (electronic supplementary material, figure S2). Two thermal regimes were established by passing mixed water through one of two conditioning units. One created a cool treatment (both populations experienced the same seasonally varying natural regime mimicking temperatures in the west of Ireland). The second unit created a warm treatment: $1.8°C \pm 0.55$ (s.d.) above the cool treatment. The cool treatment ranged from 5.9 to 16.4°C (mean = $10.8°C \pm 3.3$ s.d.) and the warm treatment ranged from 7.5 to 18.2°C (mean = $12.6°C \pm 3.4$ s.d.). The 1.8°C elevation in the warm treatment was chosen to reflect increases of 1–3°C projected under climate change scenarios [50], but was within sub-lethal ranges for brown trout [51]. The temperature was increased by 0.5°C per week when initiating treatments to minimize stress. Within each tank, 24–26 fish were lightly anaesthetized with MS-222 and marked with a unique colour combination of visible implant elastomer (VIE) tags (Northwest Marine Technology, USA) to allow for re-identification. Seven individuals lost VIE tags during the experiment, leaving $n = 95$ individually identifiable fish.

## (c) Data collection

To calculate growth rates of VIE tagged individuals across the study period, the fork length (mm) and mass (g) of lightly anaesthetized fish was recorded in April, June, July, September, and November (2017), and in April 2018 (when the study ended), with a subset of fish also measured in February 2018 during respirometry (see below) (electronic supplementary material, figure S2). We estimated growth rates as the specific growth rate (% day$^{-1}$) in terms of fork length ($G_L$) between measurement periods according to

$$G_L = 100 \times \frac{\ln S_t - \ln S_i}{d},$$

where $S_t$ is the fork length at time $t$, $S_i$ is the initial fork length, and $d$ is the time elapsed (in days) between $S_i$ and $S_t$ [52]. In February 2018 (approx. 13 months after temperature treatments were established), we measured metabolic traits in 16 fish from each temperature treatment (eight from each population, $n = 32$ individuals in total).

## (d) Measurement of SMR and MMR

The SMR of fasted individuals was determined using intermittent-flow respirometry, as described in Archer *et al.* [53] and following best practices outlined in Svendsen *et al.* [54]. SMR (mg O$_2$ h$^{-1}$) was calculated from whole-animal oxygen consumption ($\dot{M}O_2$) measurements taken overnight in a darkened controlled

temperature (CT) chamber maintained at $7.9°C \pm 0.1$ s.d. (the mid-point between the cool and warm temperature treatments at the time of measurements).

Whole-animal oxygen consumption ($\dot{M}O_2$) in animals operating at their maximum aerobic metabolic rate was used as a proxy for MMR (mg O$_2$ h$^{-1}$) [5], following best practices outlined in Norin & Clark [9]. We used an exhaustive chase protocol detailed in Archer *et al.* [53] to elicit MMR in the same individuals measured for SMR. See electronic supplementary material for a detailed description of respirometry set-up and estimation of SMR and MMR.

We calculated individual absolute AS (mg O$_2$ h$^{-1}$) as the difference between MMR and SMR.

## (e) Statistical analysis

We explored how thermal regime influenced mean values for metabolic traits (Aim 1) using two linear models (normal errors). One model included log$_{10}$SMR as the response variable, and the second included log$_{10}$MMR as the response. We included log$_{10}$body mass (at time of respirometry) as a covariate because metabolic rates are mass dependent. Both models included temperature treatment, population background, and a two-way interaction between log$_{10}$body mass and temperature treatment as explanatory variables. We calculated effect sizes as Cohen's $f$, with 95% confidence intervals (CIs) constructed by bootstrapped resampling for 10 000 resamples.

We explored how temperature treatment and metabolic phenotype influenced specific growth rates across the study period (Aim 2) within a mixed-effects modelling framework using the *nlme* package [55]. We first built a mixed-effects model (normal errors) to examine how thermal regime influenced individual-level growth rates across all study fish. The model included fixed effects of temperature treatment, population background, and time (continuous variable corresponding to months since the start of the experiment, fitted as a third-order polynomial to account for the nonlinearity of growth through time). We included a temperature treatment × time interaction to test whether thermal regime effects varied across the study. Individual identity was included as a random effect to account for multiple measurements of individuals. Since growth rate is size dependent [56], we included initial fork length as a covariate in the models. We accounted for temporal autocorrelation of growth rates by modelling an autoregressive error structure as a first-order lag function of time.

We next tested how SMR and MMR influenced growth rate patterns across temperature treatments in the subset of fish that underwent respirometry trials, using a similar modelling framework. We used the residuals of the linear relationships between each metabolic trait and body mass (all log$_{10}$-transformed) (i.e. rSMR, rMMR, electronic supplementary material, table S2 and figure S2) to correct for body mass effects on metabolic traits. The model was as described above for the first mixed-effects model but included additional fixed effects of rSMR and rMMR, and two-way interactions between rSMR and temperature treatment, and between rMMR and temperature treatment, to test if the effects of each metabolic trait on growth depended on the thermal regime. We also included the two-way interaction between rSMR and rMMR to test whether growth depended on an individual's combination of rSMR and rMMR (i.e. its composite metabolic phenotype). To explore whether the effects of SMR and MMR on growth were reflected by AS, we built another model that examined solely AS effects on growth (given that AS is determined by both SMR and MMR and is correlated with both traits: rAS and rSMR: Pearson's $r = 0.431$, $p = 0.014$; rAS and rMMR: $r = 0.995$, $p < 0.001$). This model included rAS, temperature treatment and a two-way interaction between rAS and temperature treatment to test whether growth effects of rAS depended on the thermal regime. To test if metabolic rate effects were consistent across the experiment, we constructed two additional mixed-effects models

**Table 1.** Parameter estimates and associated standard errors (s.e.), $t$-values, and $p$-values from the general linear models testing the effects of temperature treatment (cool or warm) and population background (anadromous or non-anadromous) on SMR and MMR in brown trout. SMR and MMR were $\log_{10}$-transformed, and $\log_{10}$ body mass was included as a covariate. Significance was assessed at $p < 0.05$. Effects are contrasted against fish from the anadromous population background in the cool temperature treatment.

| response | parameter | estimate | s.e. | $t$-value | $p$-value |
|---|---|---|---|---|---|
| $\log_{10}$ SMR | intercept | −1.045 | 0.300 | −3.479 | 0.002 |
| | $\log_{10}$ body mass | 0.875 | 0.144 | 6.092 | <0.001 |
| | temperature: warm | −0.078 | 0.025 | −3.090 | 0.004 |
| | population: non-anadromous | −0.061 | 0.027 | −2.246 | 0.033 |
| $\log_{10}$ MMR | intercept | 0.605 | 0.356 | 1.698 | 0.101 |
| | $\log_{10}$ body mass | 0.531 | 0.170 | 3.117 | 0.004 |
| | temperature: warm | −0.061 | 0.030 | −2.024 | 0.053 |
| | population: non-anadromous | −0.104 | 0.032 | −3.255 | 0.003 |

as described above, with additional interactions between metabolic rates and time.

Lastly, we explored whether thermal regime influenced the relationships between metabolic traits (Aim 3). We used standardized major axis regression for this analysis using the *smatr* package [57] because we had no *a priori* expectations as to which metabolic trait should drive the other (i.e. rather than predicting MMR from SMR or *vice versa*, we assumed the relationship could be symmetric, where either variable could be on either axis).

We used likelihood ratio tests (LRT) to assess statistical significance of predictor variables for all models ($\alpha = 0.05$). Non-significant interaction terms were excluded to interpret main effects. Marginal $R^2$ values for mixed-effects models were calculated using the *MuMIn* package [58]. Analysis was carried out in R v. 4.0.4 [59] and all models were checked against assumptions of the given model (independence, non-normality of residuals, heteroscedasticity, and multicollinearity).

## 3. Results

### (a) Metabolic variation

SMR and MMR were positively related to body mass as expected (table 1). We detected a significant main effect of thermal regime on SMR ($F = 9.55$, d.f. = 1, $p = 0.004$), whereby fish in the warm treatment had lower SMR (figure 1a,b), with no interaction between body mass and temperature ($F = 0.43$, d.f. = 1, $p = 0.517$). We did not detect main effects of temperature treatment on MMR ($F = 4.10$, d.f. = 1, $p = 0.053$) nor an interaction between temperature and body mass on MMR ($F = 0.06$, d.f. = 1, $p = 0.813$). There was a significant effect of population background on both SMR ($F = 5.05$, d.f. = 1, $p = 0.033$) and MMR ($F = 10.59$, d.f. = 1, $p = 0.003$; figure 1c,d). Overall, fish from the anadromous-background population tended to have higher SMR and MMR (table 1).

### (b) Metabolic traits and growth

The mixed-effects model describing specific growth rates of all fish (marginal $R^2 = 0.58$) indicated effects of thermal regime varied across the study period (LRT test for temperature treatment × time = 37.30, d.f. = 3, $p < 0.001$), with negative effects of initial size ($\chi^2 = 5.36$, d.f. = 1, $p = 0.021$) and no main effect of population ($\chi^2 = 0.44$, d.f. = 1, $p = 0.506$). The mixed-effects model that included the effects of rSMR and rMMR on growth rate (marginal $R^2 = 0.70$) retained significant two-way interactions between rSMR and temperature treatment,

between rMMR and temperature treatment and between temperature treatment and time (table 2). The effect of thermal regime on growth rates was variable through time (figure 2a). The negative rSMR × temperature treatment term indicated that in the cool treatment, higher rSMR was associated with higher growth rates, while in the warm treatment, higher rSMR was associated with lower growth rates (figure 2a,b). The positive rMMR × temperature treatment term indicated that rMMR was negatively related to growth in the cool treatments, but positively related to growth in the warm treatments (figure 2a, c). Initial body size had negative effects on growth ($\chi^2 = 10.61$, d.f. = 1, $p = 0.001$; electronic supplementary material, table S4), with no main effect of population ($\chi^2 = 0.92$, d.f. = 1, $p = 0.336$). Our additional analyses indicated that the association between both SMR and MMR and growth rate varied through time (electronic supplementary material, figure S4 and table S5) (rSMR × time LRT = 24.12, d.f. = 3, $p < 0.001$; rMMR × time LRT = 6.93, d.f. = 3, $p = 0.031$). Accounting for these interactions did not change our conclusions regarding the effects of rSMR and rMMR on growth (electronic supplementary material, table S5).

The model describing the effects of rAS on specific growth rate (marginal $R^2 = 0.67$) did not retain a significant rAS × temperature treatment interaction (table 2), and the main effects of rAS ($\chi^2 = 1.25$, d.f. = 1, $p = 0.263$), temperature treatment ($\chi^2 = 0.37$, d.f. = 1, $p = 0.542$), and population ($\chi^2 = 1.25$, d.f. = 1, $P = 0.263$) were all non-significant (figure 2a; electronic supplementary material, table S4).

### (c) Relationships between metabolic traits

There was a weak, non-significant coupling of rMMR and rSMR in both thermal regimes ($p = 0.068$ and $p = 0.100$ in cool and warm, respectively), with no effect of thermal regime on the slope ($\chi^2 = 2.00$, d.f. = 1, $p = 0.157$) or intercept of the relationship ($\chi^2 = 0.834$, $p = 0.361$) (electronic supplementary material, figure S5).

## 4. Discussion

Growth is a key performance trait linking metabolic variation to fitness via life histories [60]. Context-specific relationships between metabolic traits and growth could therefore generate fluctuating selection, which in turn could contribute to the evolutionary maintenance of spatio-temporal variation in metabolic traits [13,15,16]. Here, we provide experimental

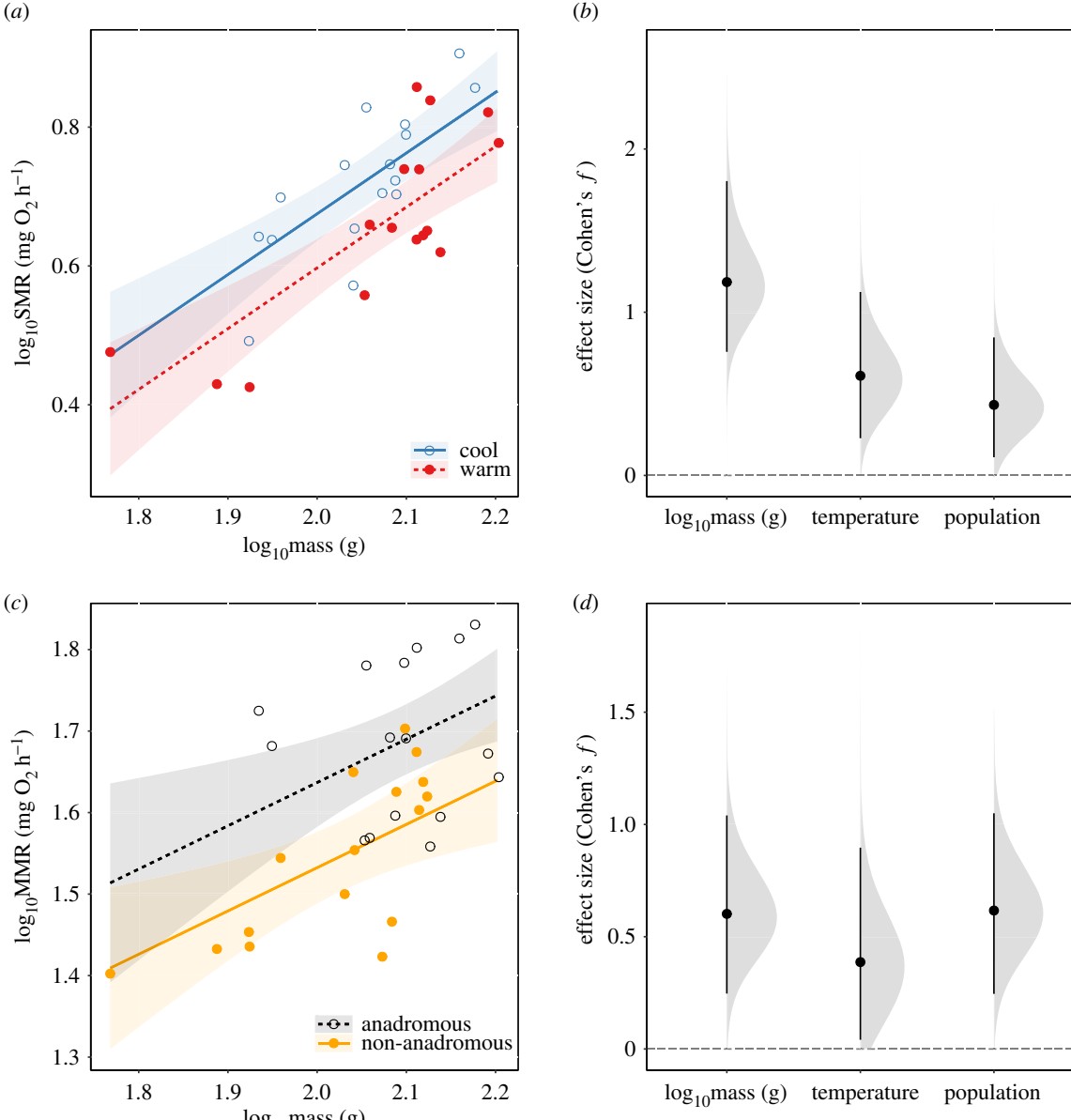

**Figure 1.** (*a*) Predicted response of standard metabolic rate (SMR, $\log_{10}$-transformed) and 95% CIs (shaded regions) to effects of $\log_{10}$body mass (g) and temperature treatment (warm or cool) and (*b*) effect size of explanatory variables on SMR. (*c*) Predicted response of maximum metabolic rate (MMR, $\log_{10}$-transformed) and 95% CIs to effects of $\log_{10}$body mass (g) and population background (anadromous or non-anadromous) and (*d*) effect size of explanatory variables on MMR. Effect sizes in (*b*) and (*d*) are shown as Cohen's *f*, together with distributions (shaded curves) and 95% CIs (black bars) obtained from non-parametric bootstrap resampling (10 000 resamples). (Online version in colour.)

evidence for relationships between two metabolic traits (SMR and MMR) and growth that depended on thermal regime. These context dependencies were in opposite directions, whereby SMR was positively linked to growth under a natural thermal regime and negatively linked under warmer temperatures that simulated climate change projections, while the opposite was true for MMR (negative effect on growth in cool, positive in warm). We also found that SMR, in turn, was lower in the warm thermal regime, consistent with an adaptive thermal acclimation response in basal metabolism that increases growth rate, and hence potentially total fitness, under persistently warmer conditions.

## (a) Implications of metabolic rate variation for growth

The positive relationship we observed between SMR and growth at cool temperatures is in agreement with previous work documenting positive associations between fitness

correlates and SMR [31–34] offering further evidence for the 'increased intake' hypothesis [2,26]—at least under cool conditions. However, the reversal of this relationship at higher temperatures supports a context dependency to the fitness consequences of a given SMR [2] and indicates that the 'compensation' hypothesis may apply in warmer environments. While less attention has been paid to links between MMR and fitness components [61], our results indicate that the fitness consequences of MMR are also context dependent, but in ways that differ from SMR. The negative MMR–growth association we observed at cool temperatures might arise from trade-offs between growth and maximum metabolic capacity, perhaps due to the expensive metabolic maintenance costs associated with digestive machinery that fuels growth [62]. At warm temperatures, such limitations might be overcome through positive effects of MMR on food consumption rates [63].

We note, however, that growth rate is just one component of fitness, and might not always map positively or linearly onto

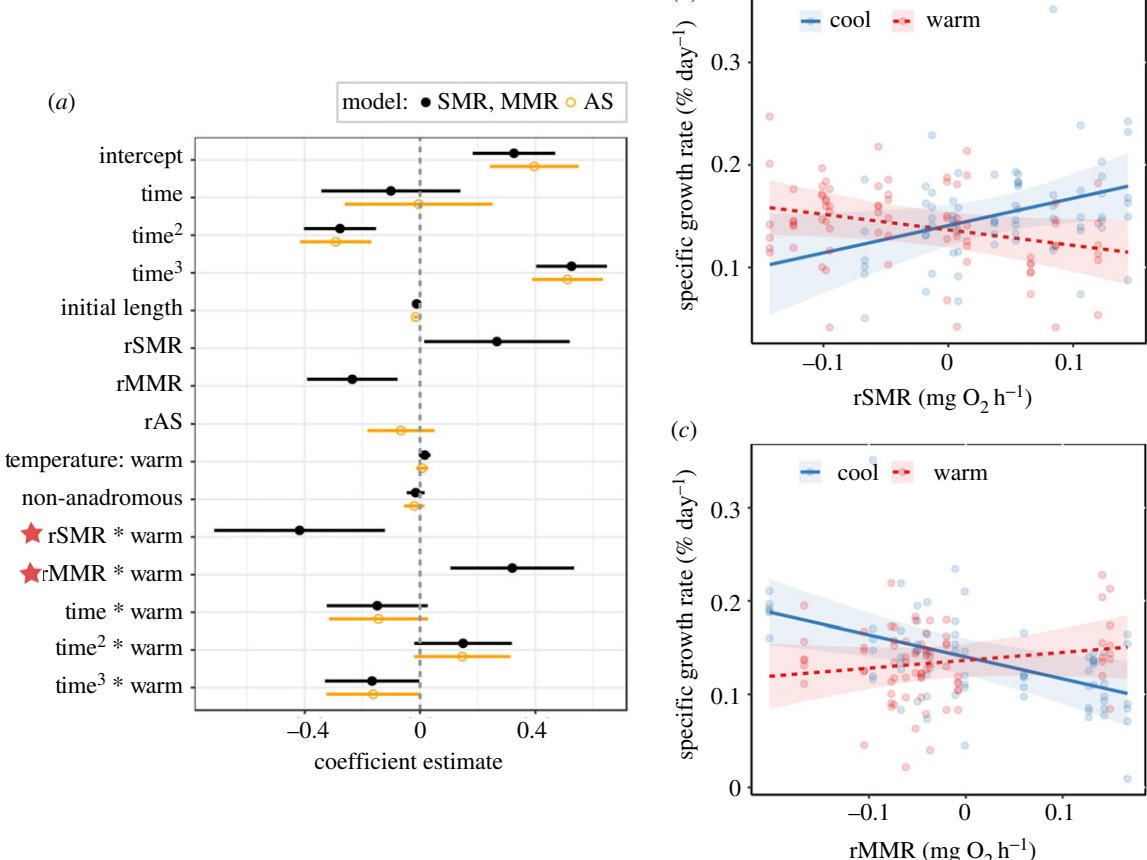

**Figure 2.** (*a*) Coefficient estimates (±95% CIs) from the mixed-effects models describing the effects of residual standard metabolic rate (rSMR), maximum metabolic rate (rMMR), aerobic scope (rAS), and temperature treatment (cool and warm) on specific growth rates of brown trout from two populations (anadromous and non-anadromous) across the study ('time' = months since initiating treatments). Interactive effects of thermal regime on metabolic rates are highlighted by stars. (*b*) Predicted growth rates in response to marginal effects of rSMR and (*c*) rMMR, at each thermal regime (shaded regions show the 95% CIs for the predictions). Growth rates were predicted at mean values for the remaining explanatory variables. (Online version in colour.)

**Table 2.** Results of the mixed-effects model analysis for specific growth rate trajectories (% day$^{-1}$) of brown trout as a function of SMR, MMR, and AS. The results of the model selection procedure on interaction terms are given (significance assessed at $p < 0.05$), and the selected model is highlighted in italics. The models included a random effect of individual identity and a first-order autoregressive correlation structure with respect to time was modelled ('time' = months since the beginning of treatment).

| excluded term | model | d.f. | AIC | logLik | *L*-ratio | *p*-value |
|---|---|---|---|---|---|---|
| | rSMR × temperature + rMMR × temperature + rSMR × rMMR + population + temperature × poly(time, 3) + length | 18 | −492.71 | 264.36 | | |
| rSMR × rMMR | *rSMR × temperature + rMMR × temperature + population + temperature × poly(time, 3) + length* | 17 | −494.69 | 264.34 | 0.03 | 0.866 |
| rSMR × temperature | rMMR × temperature + rSMR × rMMR + population + temperature × poly(time, 3) + length | 17 | −485.69 | 259.85 | 9.02 | 0.003 |
| rMMR × temperature | rSMR × temperature + rSMR × rMMR + population + temperature × poly(time, 3) + length | 17 | −485.06 | 259.53 | 9.65 | 0.002 |
| temperature × poly(time, 3) | rSMR × temperature + rMMR × temperature + rSMR × rMMR + population + poly(time, 3) + length | 15 | −488.13 | 259.06 | 10.59 | 0.014 |
| | rAS × temperature + population + temperature × poly(time, 3) + length | 15 | −490.03 | 260.02 | | |
| rAS × temperature | *rAS + temperature + population + temperature × poly(time, 3) + length* | 14 | −488.46 | 258.30 | 3.43 | 0.064 |
| temperature × poly(time, 3) | rAS × temperature + population + poly(time, 3) + length | 12 | −485.87 | 254.94 | 10.16 | 0.017 |

fitness [64]. In our study, lower SMR under warm conditions was likely beneficial for growth because maintenance energy costs were reduced, analogous to reductions in SMR facilitating higher growth or energy storage when food is limited [19]. Additional growth benefits may be provided via positive MMR–growth relationships in warmer environments, potentially stemming from enhanced competitive ability and food intake [63,65].

Temporal variation in the effects of thermal regime on growth suggested that the influence of temperature is strongest when fish are close to, or exceeding, their thermal growth limits (approx. 18°C) [66]. During the summer months of our study, rearing temperatures in the warm treatment regularly exceeded the thermal growth optimum of approximately 13.9°C for brown trout [67,68] and at times exceeded thermal growth limits, thus constraining growth. The negative SMR–growth association in the warm regime may have also served to limit growth during this period when fish in the warm treatment were pushed higher above their thermal optimum than those in the cool. Moreover, the benefits of a high metabolic rate for growth may have been limited in periods when growth potential is lower (e.g. in winter months when temperatures are sub-optimal and food availability reduced) but could have been most beneficial in the spring, when temperatures are closer to optimal ranges [67,68]. Consideration of life history might also be important here because metabolic rate tends to be positively related to migratory propensity/timing in salmonids, with individuals showing compensatory growth in the period prior to smolting [69]. In general, metabolic phenotype, growth rates, and life-history traits likely coevolve in relation to abiotic and biotic drivers that vary across space or through time [60,70].

However, it is important to note that we measured metabolic traits at a single point in time, which may not fully reflect temporal trends or within-individual variation in metabolism that could alter metabolic–growth rate associations. While relative metabolic rates among individual salmonids tend to be stable through time [71], absolute metabolic rates can vary considerably depending on a suite of factors [9,61]. In the wild, the links between metabolic traits and growth (and their context dependence) may be further modulated by additional factors, e.g. food supply [72], which in turn show spatio-temporal variability [29,35]. A natural extension to our study would be to explore metabolism–growth relationships measured at multiple timepoints and under varying conditions to test whether such links are temporally stable.

The fitness consequences of metabolic traits may also depend on complex links between the larger metabolic phenotype, i.e. the coupling (or lack thereof) between SMR and MMR, and the environmental conditions encountered [73]. In our case, we found no strong evidence for SMR–MMR coupling, nor did we find interactive effects of SMR and MMR on growth. Associations among SMR, MMR, growth, and total fitness may be different in the wild, or indeed may vary across ecological contexts [27,61]. While SMR and MMR often influence fitness via effects on AS [9,73], that growth was independent of AS in our study underscores how the effect of the composite metabolic phenotype may be obscured by context dependencies in the underlying metabolic traits. The opposing nature of the context-dependent associations that we observed between SMR and MMR on growth appeared to negate overall effects of AS on growth, suggesting that variation in metabolic traits (i.e. SMR and MMR) may be favoured

under different environmental conditions independent of their effects on AS.

## (b) Thermal regime effects on metabolic traits

The lower SMR in both populations in the warm treatments suggests that adjustment of this key physiological trait is a plastic, or acclimation, response to chronic warming. While acute warming is well known to cause an initial increase in ectotherm metabolic rates [14], exposure over longer timescales (i.e. those comparable to the more than one-year of warming in our study) tends to reduce the magnitude of the response because acclimation occurs [8]. The reduction in SMR we observed supports the potential for thermal compensation by way of the 'plastic floors' hypothesis, where lower SMR at warmer temperatures is beneficial because of reduced maintenance costs [15,22,74]. Any individual variation in thermal acclimation could translate into further growth variation via energy-saving mechanisms, with acclimation capacity generally linked to increased resilience to environmental change [8]. However, while temperatures are broadly projected to increase, more extreme and frequent warming events are also forecast [75]. The fitness consequences of a given flexible response will thus depend on both the pattern of fluctuations in temperatures and the speed at which individuals can alter their phenotype [74,76].

Intriguingly, while fish from both populations showed similar reductions in SMR in the warm regime, we detected little response in MMR. Relatively little is known about the response of MMR (and, consequently AS) to chronic temperature increases [7,21], but our results support mounting evidence indicating that many fish species show minimal warming-induced changes in MMR [77], which may potentially be explained by canalization or buffering of this key fitness-related trait. Additionally, since fish infrequently operate at MMR [78], the costs of maintaining a high MMR at warm temperatures are likely small relative to SMR, which is an unavoidable cost of living and thus potentially subject to stronger selection with warming. We note, however, that effects of long-term warming may vary considerably between, and potentially within, species [15,74]. Moreover, divergent acclimation responses of MMR compared to SMR, along with contrasting links between SMR, MMR, AS, and growth, suggest that different proximate and ultimate processes shape each metabolic component, resulting in the decoupling of metabolic traits [5,11].

## (c) Implications and considerations

Aquatic species are in widespread decline due to progressive warming and global change [79]. A better understanding of context-dependent linkages among metabolism, growth, life history, and ultimately fitness should help to inform management and conservation. For example, intraspecific variation in metabolic and growth responses to temperature could contribute to portfolio effects that foster resilience of fish stocks to climate change [80]. Our findings hint at intriguing differences in the acclimation capacity of metabolic traits that might influence the capacity to respond to environmental change, warranting further study [12,61].

Here, we measured metabolic traits after a relatively long period of temperature acclimation and growth. Repeated metabolic measurements at finer temporal scales would illuminate how metabolic phenotypes can vary according to fluctuating extrinsic and intrinsic conditions [81], and allow exploration of variation in acclimation rate, a likely important trait for

ectotherms experiencing both chronic and variable temperature changes [74]. Extending this study to include more realistic/natural conditions (e.g. co-occurring abiotic or biotic stressors) and additional populations, coupled with quantification of individual reproductive success, would give further insight into how optimal combinations of metabolic traits and life history are shaped by environmental context.

Ethics. The study and all associated procedures were carried out with ethical approval from Health Products Regulatory Authority (HPRA) Ireland, under HPRA project license AE19130/P034, and individual licenses AE19130/1087, AE19130/I200, AE19130/I201, and AE19130/I202 with all fish humanely euthanized in April 2018.

Data accessibility. The data supporting the results of this study and the R code to reproduce the analysis are available from the Dryad Digital Repository: https://doi.org/10.5061/dryad.jsxksn087 [82].

Authors' contributions. L.C.A.: conceptualization, data curation, formal analysis, investigation, methodology, project administration, validation, visualization, writing-original draft, writing-review, and editing; S.A.H.: conceptualization, investigation, methodology, project administration, writing-review, and editing; L.H.: conceptualization, investigation, methodology, project administration, writing-review, and editing; R.P.: project administration, resources, writing-review,

and editing; P.G.: project administration, resources, writing-review, and editing; P.M.: conceptualization, funding acquisition, project administration, resources, supervision, writing-review, and editing; T.E.R.: conceptualization, funding acquisition, investigation, methodology, project administration, resources, supervision, writing-review, and editing

All authors gave final approval for publication and agreed to be held accountable for the work performed therein.

Competing interests. We declare we have no competing interests

Funding. This research was supported by an ERC Starting Grant (grant no. 639192-ALH) and an SFI ERC Support Award awarded to T.E.R. P.M.c.G. was supported in part by grants from Science Foundation Ireland (grant nos. 15/IA/3028 and 16/BBSRC/3316) and by grant-in-aid (grant no. RESPI/FS/16/01) from the Marine Institute (Ireland) as part of the Marine Research Programme by the Irish Government.

Acknowledgements. The authors would like to thank Brian Clarke, Deirdre Cotter, members of the FishEyE team at UCC (particularly Robert Wynne, Ronan O'Sullivan, Peter Moran, and Adam Kane), and the staff of Inland Fisheries Ireland and the Marine Institute for obtaining brood stock and for assistance in fish rearing and husbandry. The authors are grateful to Prof. Gary Carvalho, Prof. Neil Metcalfe, and an anonymous reviewer who provided helpful comments that improved the manuscript.

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
