## [Peer Review File · Proceedings of the Royal Society B: Biological Sciences]

Review History

RSPB-2021-0074.R0 (Original submission)

Review form: Reviewer 1 (Neil Metcalfe)

Recommendation

Accept with minor revision (please list in comments)

Scientific importance: Is the manuscript an original and important contribution to its field?

Good

General interest: Is the paper of sufficient general interest?

Good

Quality of the paper: Is the overall quality of the paper suitable?

Acceptable

Is the length of the paper justified?

Yes

Should the paper be seen by a specialist statistical reviewer?

No

Do you have any concerns about statistical analyses in this paper? If so, please specify them explicitly in your report.

No

It is a condition of publication that authors make their supporting data, code and materials available - either as supplementary material or hosted in an external repository. Please rate, if applicable, the supporting data on the following criteria.

Is it accessible?

No

Is it clear?

N/A

Is it adequate?

N/A

Do you have any ethical concerns with this paper?

No

Comments to the Author

There has been much interest in recent years in the consequences of individual variation in metabolic rates, with various experimental and correlational studies attempting to test whether there is such a thing as an optimal minimal or maximal rate of metabolism. This study takes on this same theme, but provides a new angle: by comparing the growth performance of fish subjected to different temperature regimes, it shows that the impact of both minimal and maximal metabolic rates (SMR and MMR respectively) on growth rate depend on the thermal regime, with the metabolic phenotype associated with fastest growth being different in warm and cool regimes. This is an interesting finding that helps explain the persistence of intraspecific variation in metabolic phenotypes, as does the fact that SMR and MMR show contrasting patterns (and are themselves poorly correlated).

An odd feature of the manuscript is that, while the context is temperature (and the potential problems of climate change), there is no real reference to how temperature affects the biology of the study species – no mention of its upper thermal limits (Jonsson & Jonsson 2009; Elliott & Elliott 2010), or (more importantly) how its growth rate varies with temperature (Elliott, Hurley & Fryer 1995). It is only by referring to trout having an optimal temperature for growth of around 13C that we can make sense of the finding that fish in the warm and cool treatments ended up being no different in size (see comment relating to lines 227-229 below). Therefore there needs to be some discussion of how the temperatures experienced by the trout in this experiment relate to the optimal temperature for growth.

One thing that needs to be clear throughout is that phrases such as ‘temperature-dependent relationships between two metabolic traits’ (lines 256-257) are referring to the temperatures that the fish have been experiencing for months (I would hesitate to call them the acclimation temperature, since temperature fluctuated from day to day in accordance with ambient conditions), and not the temperature at which metabolic rates were measured (which was constant, and the same for all fish). It is well known that (testing) temperature affects SMR and MMR differently, but what you are showing is something more interesting and novel, so the word ‘temperature’ needs to be less ambiguous when used in conjunction with measurements of metabolic rate.

Specific points:

Lines 6-9: The impression is given in this sentence that these relationships were only found in the warm treatment fish – the next sentences indicate that this was not the case, but readers may get

confused.

Line 75: Reference 68 is relevant here, also (Robertsen et al. 2019).

Line 83: I feel we need something more targeted here by way of predictions – how might an increase in temperature be expected to alter the relationship between SMR or MMR and growth? At present the Introduction rather reads like ‘temperature affects metabolic rate in predictable ways; metabolic rate affects growth, but this depends on resource availability; therefore we investigated whether temperature influences how metabolic rate influences growth (but don’t say what level of resource availability we used, nor do we make any predictions as to which way any effect might go)’. This makes the introduction very bland and rather frustrating. The end of the penultimate paragraph emphasises the importance of resource availability in determining whether a given metabolic rate is beneficial or not – but this point is left hanging when you then go on to describe your own experiment, and you give no clue as to what you expected to find.

Line 102: Please indicate the number of broodstock females and males used from each source population.

Lines 114-115: The wording here is very ambiguous – the first part of the sentence states that 140 fish were put in one tank (presumably leaving 3 tanks empty), while the latter part says 35 fish were put in each of the four tanks. Which is correct?

Line 116: I don’t understand the statement about rearing the populations separately ‘to prevent emergence of dominance hierarchies’. Trout form dominance hierarchies even if they are all from the same population. Moreover, are you saying that fish from one population are dominant over the other? This requires explanation.

Line 118: Clarify how the fish were fed (by hand or automatic feeder, one or more meals per day, food being spread or localised), since this has a major impact on the evenness of food distribution among fish.

Lines 126-127: I am presuming that this is a 2 x 2 design (2 populations x 2 temperatures), with one tank per treatment. Please clarify.

Lines 129-130: It is ambiguous as to whether the two catchments have the same seasonal temperature cycle (so that on a given day the two populations in the cool treatment were kept at the same temperature), or whether each population has its own tailored version of a cool and a warm temperature regime. I am guessing that the former is true (since the latter would be very complex to generate), but the wording actually implies the latter.

Lines 142-144: This is a bit confusing – growth rates were calculated ‘across the study period’, but the last measurement of size seems to have been in November 2017, which is 3 months before the measurement of metabolic rates and 5 months before the end of the study (according to line 112). I am presuming that the subset of fish subjected to respirometry must have been measured in February 2018 – maybe all were measured at the same time? Fig S3 also suggests that measurements continued until April 2018. Please clarify.

Line 183: Note the typo – should be $P < 0.001$, not > 0.001 .

Line 199: Clarify which body mass this is – the mass at the time of measuring metabolic rate?

Note that there is no mention of you having recorded body mass at any point in the experiment!

Line 195: ‘explored how temperature affected metabolic variation’ is not an ideal description of this analysis, since it could imply that you are referring to the temperature at which metabolic rate was measured (when you are actually referring to the rearing temperature, metabolic rates of all fish having been measured at the same temperature). Also it could imply that you are testing whether temperature affects the extent of variability in metabolic rates, when in fact you are testing if it affects the mean. Please re-phrase.

Line 214: The order in which the Results are presented seems slightly illogical to me. The analysis of growth rates is heavily focussed on the effect of metabolic rate, yet we are not told about what influences metabolic rate until later. It might be better to show the analyses of factors influencing metabolic rate first, and then go on to look at the consequences of this variation (i.e. for growth).

Lines 227-229: Fig S3 reveals something that could usefully be brought out: it would appear to be the case that warm treatment fish initially grew faster than cool (up until the summer), but then grew more slowly. This would make intuitive sense, since the warm treatment initially brought fish closer to the optimal temperature for growth in trout, but would then (by summer) have taken them further above it than the cool treatment. Adding temperature x time interactions to the model described here and in Table S4 would test for this. We need some exploration of this

changing pattern, because readers will otherwise be puzzled why a temperature treatment seems to have no overall effect on growth (i.e. there is no main effect of temperature in the final model presented in Table 1). Furthermore, Fig. S3 also suggests that, regardless of thermal regime, low SMR fish had faster growth rates than high SMR fish at the start of the experiment (in the spring), but this gradually reversed so that by the end of the experiment (late winter) the growth rate was markedly higher in high SMR fish than in low. This striking pattern deserves some comment/discussion – is a low SMR beneficial at times when fish are growing little, but did the pattern reverse towards the following spring as small (i.e. high SMR) fish showed compensatory growth in the period leading up to smolting, as has been repeatedly shown in presmolt salmon (Nicieza & Braña 1993; Sigourney et al. 2013; Thomson & Lyndon 2018; Simmons et al. 2020)?

Line 324: It might be worth pointing out that there is perhaps no higher cost of a high MMR when temperatures are warm, because the animal is rarely operating at its MMR and is not forced to do so. This is quite different from the situation with SMR, which is an inevitable cost of living, and one that is higher at higher temperatures, leading to strong selection to reduce it.

Lines 332-342: You start this section saying that you are cautious about making population-level inferences – then go on to make them! I feel this is very speculative given that $n=1$ for migrant vs. resident populations, especially since the small pool of adults used to create the experimental populations. I suggest dropping this section.

Lines 634-635: The acronyms 'AB' and 'Non-AB' do not feature in the Figure, so are not needed here.

Supplementary information: Please expand on the description of how MMR was calculated – the rate of oxygen depletion would presumably take a few seconds to increase once the fish had been placed in the respirometer (since it takes that time for water to circulate around the closed system), so did you exclude the first few readings, or only take the highest readings?

Table S1: The sample size ($n=32$) is ambiguous – does this refer to the number from each population, or in total? It would be better to give the sample size for each treatment group. Were these sizes representative of the fish in each treatment, or were they selected to maximise the size overlap between treatments? If the former, then it is odd that there is no size difference between fish reared in warm and cool treatments

I hope that these comments are helpful

Neil Metcalfe (I am happy to be identified to the authors)

Elliott, J.M. & Elliott, J.A. (2010) Temperature requirements of Atlantic salmon *Salmo salar*, brown trout *Salmo trutta* and Arctic charr *Salvelinus alpinus*: predicting the effects of climate change. *Journal of Fish Biology*, 77, 1793-1817.

Elliott, J.M., Hurley, M.A. & Fryer, R.J. (1995) A new improved growth model for brown trout, *Salmo trutta*. *Functional Ecology*, 9, 290-298.

Jonsson, B. & Jonsson, N. (2009) A review of the likely effects of climate change on anadromous Atlantic salmon *Salmo salar* and brown trout *Salmo trutta*, with particular reference to water temperature and flow. *Journal of Fish Biology*, 75, 2381-2447.

Nicieza, A.G. & Braña, F. (1993) Compensatory growth and optimum size in one-year-old smolts of Atlantic salmon (*Salmo salar*). *Production of juvenile salmon, Salmo salar*, in natural waters (eds R.J. Gibson & R.E. Cutting), pp. 225-237.

Robertsen, G., Reid, D., Einum, S., Aronsen, T., Fleming, I.A., Sundt-Hansen, L.E., Karlsson, S., Kvingedal, E., Ugedal, O. & Hindar, K. (2019) Can variation in standard metabolic rate explain context-dependent performance of farmed Atlantic salmon offspring? *Ecology and Evolution*, 9, 212-222.

Sigourney, D.B., Letcher, B.H., Obedzinski, M. & Cunjak, R.A. (2013) Interactive effects of life history and season on size-dependent growth in juvenile Atlantic salmon. *Ecology of Freshwater Fish*, 22, 495-507.

Simmons, O.M., Britton, J.R., Gillingham, P.K. & Gregory, S.D. (2020) Influence of environmental and biological factors on the overwinter growth rate of Atlantic salmon *Salmo salar* parr in a UK chalk stream. *Ecology of Freshwater Fish*, 29, 665-678.

Thomson, M. & Lyndon, A.R. (2018) Comparing anadromous brown trout *Salmo trutta* in small,

neighbouring catchments across contrasting landscapes: What is the role of environment in determining life-history characteristics? *Journal of Fish Biology*, 92, 593-606.

Review form: Reviewer 2

Recommendation

Accept with minor revision (please list in comments)

Scientific importance: Is the manuscript an original and important contribution to its field?

Good

General interest: Is the paper of sufficient general interest?

Good

Quality of the paper: Is the overall quality of the paper suitable?

Good

Is the length of the paper justified?

Yes

Should the paper be seen by a specialist statistical reviewer?

No

Do you have any concerns about statistical analyses in this paper? If so, please specify them explicitly in your report.

No

It is a condition of publication that authors make their supporting data, code and materials available - either as supplementary material or hosted in an external repository. Please rate, if applicable, the supporting data on the following criteria.

Is it accessible?

Yes

Is it clear?

Yes

Is it adequate?

Yes

Do you have any ethical concerns with this paper?

No

Comments to the Author

See attached documents. (See Appendix A)

Decision letter (RSPB-2021-0074.R0)

18-Feb-2021

Dear Dr Archer:

I am writing to inform you that your manuscript RSPB-2021-0074 entitled "Associations between metabolic traits and growth rate are temperature dependent in brown trout (Salmo trutta)" has, in its current form, been rejected for publication in Proceedings B.

This action has been taken on the advice of referees, who have recommended that substantial revisions are necessary. With this in mind we would be happy to consider a resubmission, provided the comments of the referees are fully addressed. However please note that this is not a provisional acceptance.

Sincerely,
Professor Gary Carvalho
mailto: proceedingsb@royalsociety.org

Associate Editor
Comments to Author:

Your manuscript has now been reviewed by two expert reviewers.

Both reviewers found the manuscript to be interesting, and the results to potentially represent a significant contribution to the general topic of metabolism in the face of warming. However, both reviewers also pointed out a number of issues related to methods that were unclear. And in places, they made specific suggestions for how to make your arguments more clear and/or compelling.

In addition, the first reviewer raised two additional points: First, he (Metcalf) recommends a change to one of the statistical tests. And second, he rightly points out that you don't include sufficient background on the thermal biology of the study species.

To conclude, the two reviewers clearly saw significant value in your study (as do I). For this reason, I speculate, they both provided detailed reviews that I hope you find helpful in improving the manuscript.

Reviewer(s)' Comments to Author:

Referee: 1

Comments to the Author(s)

There has been much interest in recent years in the consequences of individual variation in metabolic rates, with various experimental and correlational studies attempting to test whether there is such a thing as an optimal minimal or maximal rate of metabolism. This study takes on this same theme, but provides a new angle: by comparing the growth performance of fish subjected to different temperature regimes, it shows that the impact of both minimal and maximal metabolic rates (SMR and MMR respectively) on growth rate depend on the thermal regime, with the metabolic phenotype associated with fastest growth being different in warm and cool regimes. This is an interesting finding that helps explain the persistence of intraspecific variation in metabolic phenotypes, as does the fact that SMR and MMR show contrasting patterns (and are themselves poorly correlated).

An odd feature of the manuscript is that, while the context is temperature (and the potential problems of climate change), there is no real reference to how temperature affects the biology of the study species – no mention of its upper thermal limits (Jonsson & Jonsson 2009; Elliott & Elliott 2010), or (more importantly) how its growth rate varies with temperature (Elliott, Hurley & Fryer 1995). It is only by referring to trout having an optimal temperature for growth of around 13°C that we can make sense of the finding that fish in the warm and cool treatments ended up being no different in size (see comment relating to lines 227-229 below). Therefore there needs to be some discussion of how the temperatures experienced by the trout in this experiment relate to the optimal temperature for growth.

One thing that needs to be clear throughout is that phrases such as 'temperature-dependent relationships between two metabolic traits' (lines 256-257) are referring to the temperatures that the fish have been experiencing for months (I would hesitate to call them the acclimation temperature, since temperature fluctuated from day to day in accordance with ambient conditions), and not the temperature at which metabolic rates were measured (which was constant, and the same for all fish). It is well known that (testing) temperature affects SMR and MMR differently, but what you are showing is something more interesting and novel, so the word 'temperature' needs to be less ambiguous when used in conjunction with measurements of metabolic rate.

Specific points:

Lines 6-9: The impression is given in this sentence that these relationships were only found in the warm treatment fish – the next sentences indicate that this was not the case, but readers may get confused.

Line 75: Reference 68 is relevant here, also (Robertson et al. 2019).

Line 83: I feel we need something more targeted here by way of predictions – how might an increase in temperature be expected to alter the relationship between SMR or MMR and growth? At present the Introduction rather reads like 'temperature affects metabolic rate in predictable ways; metabolic rate affects growth, but this depends on resource availability; therefore we investigated whether temperature influences how metabolic rate influences growth (but don't say what level of resource availability we used, nor do we make any predictions as to which way any effect might go)'. This makes the introduction very bland and rather frustrating. The end of the penultimate paragraph emphasises the importance of resource availability in determining whether a given metabolic rate is beneficial or not – but this point is left hanging when you then go on to describe your own experiment, and you give no clue as to what you expected to find.

Line 102: Please indicate the number of broodstock females and males used from each source population.

Lines 114-115: The wording here is very ambiguous – the first part of the sentence states that 140 fish were put in one tank (presumably leaving 3 tanks empty), while the latter part says 35 fish were put in each of the four tanks. Which is correct?

Line 116: I don't understand the statement about rearing the populations separately 'to prevent emergence of dominance hierarchies'. Trout form dominance hierarchies even if they are all from the same population. Moreover, are you saying that fish from one population are dominant over the other? This requires explanation.

Line 118: Clarify how the fish were fed (by hand or automatic feeder, one or more meals per day, food being spread or localised), since this has a major impact on the evenness of food distribution among fish.

Lines 126-127: I am presuming that this is a 2 x 2 design (2 populations x 2 temperatures), with one tank per treatment. Please clarify.

Lines 129-130: It is ambiguous as to whether the two catchments have the same seasonal temperature cycle (so that on a given day the two populations in the cool treatment were kept at the same temperature), or whether each population has its own tailored version of a cool and a warm temperature regime. I am guessing that the former is true (since the latter would be very complex to generate), but the wording actually implies the latter.

Lines 142-144: This is a bit confusing – growth rates were calculated 'across the study period', but the last measurement of size seems to have been in November 2017, which is 3 months before the measurement of metabolic rates and 5 months before the end of the study (according to line 112). I am presuming that the subset of fish subjected to respirometry must have been measured in February 2018 – maybe all were measured at the same time? Fig S3 also suggests that measurements continued until April 2018. Please clarify.

Line 183: Note the typo – should be $P < 0.001$, not > 0.001 .

Line 199: Clarify which body mass this is – the mass at the time of measuring metabolic rate?

Note that there is no mention of you having recorded body mass at any point in the experiment!

Line 195: 'explored how temperature affected metabolic variation' is not an ideal description of this analysis, since it could imply that you are referring to the temperature at which metabolic rate was measured (when you are actually referring to the rearing temperature, metabolic rates of all fish having been measured at the same temperature). Also it could imply that you are testing whether temperature affects the extent of variability in metabolic rates, when in fact you are testing if it affects the mean. Please re-phrase.

Line 214: The order in which the Results are presented seems slightly illogical to me. The analysis of growth rates is heavily focussed on the effect of metabolic rate, yet we are not told about what influences metabolic rate until later. It might be better to show the analyses of factors influencing metabolic rate first, and then go on to look at the consequences of this variation (i.e. for growth).

Lines 227-229: Fig S3 reveals something that could usefully be brought out: it would appear to be the case that warm treatment fish initially grew faster than cool (up until the summer), but then grew more slowly. This would make intuitive sense, since the warm treatment initially brought fish closer to the optimal temperature for growth in trout, but would then (by summer) have taken them further above it than the cool treatment. Adding temperature x time interactions to the model described here and in Table S4 would test for this. We need some exploration of this changing pattern, because readers will otherwise be puzzled why a temperature treatment seems to have no overall effect on growth (i.e. there is no main effect of temperature in the final model presented in Table 1). Furthermore, Fig. S3 also suggests that, regardless of thermal regime, low SMR fish had faster growth rates than high SMR fish at the start of the experiment (in the spring), but this gradually reversed so that by the end of the experiment (late winter) the growth rate was markedly higher in high SMR fish than in low. This striking pattern deserves some comment/discussion – is a low SMR beneficial at times when fish are growing little, but did the pattern reverse towards the following spring as small (i.e. high SMR) fish showed compensatory growth in the period leading up to smolting, as has been repeatedly shown in presmolt salmon (Nicieza & Braña 1993; Sigourney et al. 2013; Thomson & Lyndon 2018; Simmons et al. 2020)?

Line 324: It might be worth pointing out that there is perhaps no higher cost of a high MMR when temperatures are warm, because the animal is rarely operating at its MMR and is not forced to do so. This is quite different from the situation with SMR, which is an inevitable cost of living, and one that is higher at higher temperatures, leading to strong selection to reduce it.

Lines 332-342: You start this section saying that you are cautious about making population-level inferences – then go on to make them! I feel this is very speculative given that $n=1$ for migrant vs.

resident populations, especially since the small pool of adults used to create the experimental populations. I suggest dropping this section.

Lines 634-635: The acronyms 'AB' and 'Non-AB' do not feature in the Figure, so are not needed here.

Supplementary information: Please expand on the description of how MMR was calculated – the rate of oxygen depletion would presumably take a few seconds to increase once the fish had been placed in the respirometer (since it takes that time for water to circulate around the closed system), so did you exclude the first few readings, or only take the highest readings?

Table S1: The sample size (n=32) is ambiguous – does this refer to the number from each population, or in total? It would be better to give the sample size for each treatment group. Were these sizes representative of the fish in each treatment, or were they selected to maximise the size overlap between treatments? If the former, then it is odd that there is no size difference between fish reared in warm and cool treatments

I hope that these comments are helpful

Neil Metcalfe (I am happy to be identified to the authors)

Elliott, J.M. & Elliott, J.A. (2010) Temperature requirements of Atlantic salmon *Salmo salar*, brown trout *Salmo trutta* and Arctic charr *Salvelinus alpinus*: predicting the effects of climate change. *Journal of Fish Biology*, 77, 1793-1817.

Elliott, J.M., Hurley, M.A. & Fryer, R.J. (1995) A new improved growth model for brown trout, *Salmo trutta*. *Functional Ecology*, 9, 290-298.

Jonsson, B. & Jonsson, N. (2009) A review of the likely effects of climate change on anadromous Atlantic salmon *Salmo salar* and brown trout *Salmo trutta*, with particular reference to water temperature and flow. *Journal of Fish Biology*, 75, 2381-2447.

Nicieza, A.G. & Braña, F. (1993) Compensatory growth and optimum size in one-year-old smolts of Atlantic salmon (*Salmo salar*). *Production of juvenile salmon, Salmo salar, in natural waters* (eds R.J. Gibson & R.E. Cutting), pp. 225-237.

Robertsen, G., Reid, D., Einum, S., Aronsen, T., Fleming, I.A., Sundt-Hansen, L.E., Karlsson, S., Kvingedal, E., Ugedal, O. & Hindar, K. (2019) Can variation in standard metabolic rate explain context-dependent performance of farmed Atlantic salmon offspring? *Ecology and Evolution*, 9, 212-222.

Sigourney, D.B., Letcher, B.H., Obedzinski, M. & Cunjak, R.A. (2013) Interactive effects of life history and season on size-dependent growth in juvenile Atlantic salmon. *Ecology of Freshwater Fish*, 22, 495-507.

Simmons, O.M., Britton, J.R., Gillingham, P.K. & Gregory, S.D. (2020) Influence of environmental and biological factors on the overwinter growth rate of Atlantic salmon *Salmo salar* parr in a UK chalk stream. *Ecology of Freshwater Fish*, 29, 665-678.

Thomson, M. & Lyndon, A.R. (2018) Comparing anadromous brown trout *Salmo trutta* in small, neighbouring catchments across contrasting landscapes: What is the role of environment in determining life-history characteristics? *Journal of Fish Biology*, 92, 593-606.

Referee: 2

Comments to the Author(s)

See attached documents

Author's Response to Decision Letter for (RSPB-2021-0074.R0)

See Appendix B.

RSPB-2021-1509.R0

Review form: Reviewer 1 (Neil Metcalfe)

Recommendation

Accept with minor revision (please list in comments)

Scientific importance: Is the manuscript an original and important contribution to its field?

Good

General interest: Is the paper of sufficient general interest?

Good

Quality of the paper: Is the overall quality of the paper suitable?

Excellent

Is the length of the paper justified?

Yes

Should the paper be seen by a specialist statistical reviewer?

No

Do you have any concerns about statistical analyses in this paper? If so, please specify them explicitly in your report.

No

It is a condition of publication that authors make their supporting data, code and materials available - either as supplementary material or hosted in an external repository. Please rate, if applicable, the supporting data on the following criteria.

Is it accessible?

Yes

Is it clear?

Yes

Is it adequate?

Yes

Do you have any ethical concerns with this paper?

No

Comments to the Author

The authors have done an excellent job in responding to the points raised by myself and the other reviewer, and as a result the manuscript is much improved. I only have a small number of points, mostly relating to typos:

Line 79: Should be 'may be' not 'maybe'.

Line 209-210: Slight re-phrasing to aid clarity: '...rSMR and temperature, and between rMMR and temperature, to test if...'

Line 211: Insert 'the' - '...included the two-way interaction...'

Lines 312-318: The factual statements about trout thermal biology need to be more clearly referenced – at present it is not clear where you get the information on the thermal growth optimum nor the thermal growth limits for trout. Indeed, the two Elliott references (refs 66 and 67) are quoted as if they referred to your experiment. Also you need to clarify that the ‘thermal limit’ quoted on lines 312-313 is that for growth (trout can temporarily survive much higher temperatures than 18C).

Line 319: ‘have been’ not ‘have be’.

Line 329: Delete ‘and in a subset of fish’ – the sample size is not relevant to the arguments you are making here, which relate only to the single measurement per individual.

Neil Metcalfe (I am happy to be identified to the authors)

Review form: Reviewer 2

Recommendation

Accept as is

Scientific importance: Is the manuscript an original and important contribution to its field?

Excellent

General interest: Is the paper of sufficient general interest?

Excellent

Quality of the paper: Is the overall quality of the paper suitable?

Excellent

Is the length of the paper justified?

Yes

Should the paper be seen by a specialist statistical reviewer?

No

Do you have any concerns about statistical analyses in this paper? If so, please specify them explicitly in your report.

No

It is a condition of publication that authors make their supporting data, code and materials available - either as supplementary material or hosted in an external repository. Please rate, if applicable, the supporting data on the following criteria.

Is it accessible?

Yes

Is it clear?

Yes

Is it adequate?

Yes

Do you have any ethical concerns with this paper?

No

Comments to the Author

This was the easiest second round review I have ever done. The concerns of both original reviewers were responded to thoroughly and the manuscript seems much improved because of it. I would accept the manuscript as is.

Decision letter (RSPB-2021-1509.R0)

16-Aug-2021

Dear Dr Archer

I am pleased to inform you that your manuscript RSPB-2021-1509 entitled "Associations between metabolic traits and growth rate in brown trout (*Salmo trutta*) depend on thermal regime" has been accepted for publication in Proceedings B.

The referee(s) have recommended publication, but also suggest some minor revisions to your manuscript. Therefore, I invite you to respond to the referee(s)' comments and revise your manuscript. Because the schedule for publication is very tight, it is a condition of publication that you submit the revised version of your manuscript within 7 days. If you do not think you will be able to meet this date please let us know.

Sincerely,

Professor Gary Carvalho

Reviewer(s)' Comments to Author:

Referee: 1

Comments to the Author(s).

The authors have done an excellent job in responding to the points raised by myself and the other reviewer, and as a result the manuscript is much improved. I only have a small number of points, mostly relating to typos:

Line 79: Should be 'may be' not 'maybe'.

Line 209-210: Slight re-phrasing to aid clarity: '...rSMR and temperature, and between rMMR and temperature, to test if...'

Line 211: Insert 'the' - '...included the two-way interaction...'

Lines 312-318: The factual statements about trout thermal biology need to be more clearly referenced - at present it is not clear where you get the information on the thermal growth optimum nor the thermal growth limits for trout. Indeed, the two Elliott references (refs 66 and 67) are quoted as if they referred to your experiment. Also you need to clarify that the 'thermal limit' quoted on lines 312-313 is that for growth (trout can temporarily survive much higher temperatures than 18C).

Line 319: 'have been' not 'have be'.

Line 329: Delete 'and in a subset of fish' - the sample size is not relevant to the arguments you are making here, which relate only to the single measurement per individual.

Neil Metcalfe (I am happy to be identified to the authors)

Referee: 2

Comments to the Author(s).

This was the easiest second round review I have ever done. The concerns of both original reviewers were responded to thoroughly and the manuscript seems much improved because of it. I would accept the manuscript as is.

Author's Response to Decision Letter for (RSPB-2021-1509.R0)

See Appendix C.

Decision letter (RSPB-2021-1509.R1)

17-Aug-2021

Dear Dr Archer

I am pleased to inform you that your manuscript entitled "Associations between metabolic traits and growth rate in brown trout (*Salmo trutta*) depend on thermal regime" has been accepted for publication in Proceedings B.

Data Accessibility section

Open Access

Paper charges

Sincerely,

Proceedings B

Appendix A

Review of Archer *et al.* (2021) Proc B Soc

Associations between metabolic traits and growth rate are temperature dependent in brown trout (*Salmo trutta*)

An initial disclaimer: I do not have the expertise in fish experiments, but the study seems well designed given the constraints of experimentally working on large animals. I also do not have experience with measurements of SMR and MMR on fish. I have therefore reviewed the paper given my experience of the effects of short- and long-term warming on biological rates and the statistical analyses.

In this manuscript, the authors look at the response of brown trout to a long-term difference in temperature. This is a highly important question given the current climate change emergency that the world is undergoing. Specifically, there is a lot of evidence of marine life moving to new environments as temperatures change, but less evidence of the ability of fish to acclimate to warmer temperatures. Can fish acclimate to warmer temperatures? Here, the authors try to tackle this challenge by linking metabolism traits to growth rates (a fitness related trait). The authors run a long-term experiment (~1 year) where some fish were kept at conditions ~1.8 °C higher than others. Growth was tracked in ~100 individuals every few months and metabolic traits were measured at the end of the experiment.

The main finding is that there is an interaction between the effect of temperature and metabolic traits on growth rate. More interestingly, the effect flips between the metabolic ceiling (maximum metabolic rate) and the metabolic floor (standard metabolic rate), such that the standard metabolic rate is flexible and acclimates to optimise growth at two different statistical regimes.

I enjoyed reading the manuscript and found the results compelling. The statistical analyses are well thought out and appear robust. I am especially impressed by the presence of the data analysis code in an online repository for publication. I downloaded it and was able to reproduce the plots and analysis. I have a few major issues (see below) mainly about limitations in the dataset. However, given the robustness of the analysis I do not think these should limit its publication. I would like to congratulate them on a well written manuscript.

Major comments

- My main issue is the lack of metabolic rate measurements through time, and as such there are only 32 measurements across the entire growth rate dataset (~150). Having metabolic rate measurements through the entire experiment alongside growth rates would have of course been optimal (and the authors acknowledge this in the last paragraph of the discussion). This is especially important as the authors acknowledge that growth rate and metabolic rates are size dependent, so having a single value at the end of the experiment to link to growth rates when they were at a different size is less than ideal. Given the variability in individual growth rate through time, how much individual variation in metabolic traits do the authors think were missed by not having these measurements?
- I got a bit lost in the timescale of the study. I think the study would benefit from a schematic showing when all the measurements were taken and the number of points for each measurement.
- I got quite confused about the number of data points in the study (See minor comment Line 137). Think you need to state again on Line 171 that this is the 32 individuals fish for which metabolic trials were taken. And possibly just be more up front about the limits of the study

when talking about the analysis or in the discussion in regards to using metabolic traits from the end of the experiment to predict growth rates taken months earlier.

Minor comments

- Line 7: By metabolic traits the authors mean SMR and MMR I assume. Maybe metabolic traits can be set up in the first sentence alongside defining SMR and MMR?
- Line 20-22: Nice and dramatic opening as is usual for metabolism-related papers. :)
- Line 22: “metabolic traits” need to be defined better here. Maybe just bringing Lines 27-34 up?
- Line 28: define thermoneutral zone
- Line 50: temperature effects... on metabolism?
- Line 52: feel like there needs to be some qualifier or references to state “acute effects are reasonably well researched”
- Line 72: RMR is a typo. Should be BMR I think.
- Line 120: Are Skretting Ltd the food company? Reference at the end of the sentence feels like you’re referencing their method?
- Line 123-124: Really appreciate the transparency and honesty here.
- Line 137: Seems to be some confusion of using populations in the study.
 - For example Line 115 it says there are two tanks per population.
 - Line 137: population becomes each individual tank
 - Would suggest being really explicit about this early on
- Line 143: As well as dates it would be useful to know how long the study has been going on.
- Line 171-174: Just to be absolutely clear the raw values of SMR and MMR were regressed against body mass (all log₁₀). The values that went forward into the main analysis were the residuals of this analysis? Also were these residuals correlated (i.e. did individuals with higher residuals for MMR have higher residuals for SMR?). This information is present on Line 246 but it might be useful as a Supplementary plot.
- Line 194: If you’re always using a normal error structure, you could just say linear mixed effects models throughout, instead of GLMs.
- Lines 203 - 205: Do you have an *a priori* expectation for whether SMR or MMR should drive each other? If not, then Standardised Major Axis Regression might be used here. Granted though I have not looked at whether it can be used with covariates. *smatr* is a very well made R package to do this though.
- Line 209: Would use $p < 0.05$ than %
- Figure 1: in (a) could you highlight the key results here? The $r_{SMR} * Warm$ and $r_{MMR} * warm$ results. Shade that background a tiny bit or put stars by the factors? Would allow readers to more easily find the key result
- Figure 1: b and c. I always struggle when there are predictions and no points on the plot when it is not a theoretical model. Could you possibly put on partial residuals? These can be taken from a *visreg* object - <https://pbreheny.github.io/visreg/> - and are used to show the impact of a predictor given other covariates. This is very similar to how the predictions were created. If nothing else it will allow the reader to see how many points underpin the dataset.

Appendix B

University College Cork
Distillery Fields Campus
North Mall
Cork
Ireland

2nd July 2021

Dear Professor Carvalho,

Many thanks for your letter dated 18th February 2021, detailing the reviewers' thoughts on our manuscript: RSPB-2021-0074 "*Associations between metabolic traits and growth rate are temperature dependent in brown trout (Salmo trutta)*" for publication in *Proceedings of the Royal Society B: Biological Sciences*. We deeply appreciate the time taken to provide such thoughtful comments and excellent suggestions on how to improve our study, which we have taken on board to produce a better manuscript.

Below, we have responded, point by point, to the reviewers' comments with quotations and line numbers to our revised text. Please also find two Word documents as part of this resubmission. The first file entitled "Archer et al Metabolic traits and growth Revision (Track Changes)" contains the original manuscript and details of the track changes that we have made in the revision. The second file entitled "Archer et al Metabolic traits and growth Revision" is the revised manuscript with all track changes accepted. We hope that this version of the manuscript is now suitable for publication in *Proceedings of the Royal Society B: Biological Sciences*.

The manuscript presented here has neither been published nor is under consideration elsewhere, including the internet. Thank you for your consideration, and please do not hesitate to contact us if you have any questions.

Yours faithfully (on behalf of co-authors),

Louise Archer
Phone: +353 (0)21 490 4677
E-mail: l.archer@umail.ucc.ie

Associate Editor

Comments to Author:

Your manuscript has now been reviewed by two expert reviewers.

Both reviewers found the manuscript to be interesting, and the results to potentially represent a significant contribution to the general topic of metabolism in the face of warming. However, both reviewers also pointed out a number of issues related to methods that were unclear. And in places, they made specific suggestions for how to make your arguments more clear and/or compelling.

In addition, the first reviewer raised two additional points: First, he (Metcalf) recommends a change to one of the statistical tests. And second, he rightly points out that you don't include sufficient background on the thermal biology of the study species.

To conclude, the two reviewers clearly saw significant value in your study (as do I). For this reason, I speculate, they both provided detailed reviews that I hope you find helpful in improving the manuscript.

Response – We are grateful for the overall positive reception to our study and the detailed and thoughtful suggestions for how to improve our manuscript. We have taken all of this information on board and done our best to actively address all points. We have (1) provided additional clarity on the methods and analysis used; (2) provided some further discussion on both the context and the relevance of our study (emphasising our focus on understudied chronic temperature effects); (3) modified our analysis to incorporate the suggestion by Reviewer 1; and (4) included some additional information on the thermal biology of trout to place our findings in context. Please see below for our more detailed responses.

***Note that all line numbers refer to the revised version of the manuscripts unless otherwise stated

Referee #1: Comments to the Author(s)

There has been much interest in recent years in the consequences of individual variation in metabolic rates, with various experimental and correlational studies attempting to test whether there is such a thing as an optimal minimal or maximal rate of metabolism. This study takes on this same theme, but provides a new angle: by comparing the growth performance of fish subjected to different temperature regimes, it shows that the impact of both minimal and maximal metabolic rates (SMR and MMR respectively) on growth rate depend on the thermal regime, with the metabolic phenotype associated with fastest growth being different in warm and cool regimes. This is an interesting finding that helps explain the persistence of intraspecific variation in metabolic phenotypes, as does the fact that SMR and MMR show contrasting patterns (and are themselves poorly correlated).

Response 1.1 – Thank you for this overall positive appraisal of our study - we hope that our findings help to better understand the much-debated variation that exists among metabolic phenotypes and to stimulate further research. In our revision, we have (1) included some detail on the thermal biology of trout and how our findings fit within this context; (2) emphasised that our study addresses the effects of long-term rather than short-term/acute temperatures by referring to effects of “*thermal regime*” instead of simply “*temperature*” throughout; and (3) modified our analysis to explore the consistency of the effects of thermal regime through time. Please see our more detailed responses below and thank you for challenging us to explore the intricacies of our study in more depth.

An odd feature of the manuscript is that, while the context is temperature (and the potential problems of climate change), there is no real reference to how temperature affects the biology of the study species – no mention of its upper thermal limits (Jonsson & Jonsson 2009; Elliott & Elliott 2010), or (more importantly) how its growth rate varies with temperature (Elliott, Hurley & Fryer 1995). It is only by referring to trout having an optimal temperature for growth of around 13C that we can make sense of the finding that fish in the warm and cool treatments ended up being no different in size (see comment relating to lines 227-229 below). Therefore there needs to be some discussion of how the temperatures experienced by the trout in this experiment relate to the optimal temperature for growth.

Response 1.2 – We appreciate you drawing our attention to this omission. We completely agree that the thermal biology of brown trout is essential for placing our results in context, and we recognise that there was insufficient consideration of this point in our initial submission. We have now included some additional discussion (Ln 312 – Ln 319) on the thermal biology of brown trout in relation to our findings which we hope provides the necessary additional background detail to better understand the context and outcome of our study.

Ln 312: *“Temporal variation in the effects of thermal regime on growth suggested that the influence of temperature on growth is strongest when fish are closer to their thermal limits (~18 °C). During the summer months of our study, rearing temperatures in the warm treatment regularly exceeded the thermal growth optimum of ~13.9 °C for brown trout, and at times exceeded thermal growth limits, thus constraining growth. The negative SMR – growth association in the warm regime may have also served to limit growth during this period when fish in the warm treatment were pushed higher above their thermal optimum than those in the cool (66, 67).”*

One thing that needs to be clear throughout is that phrases such as ‘temperature-dependent relationships between two metabolic traits’ (lines 256-257) are referring to the temperatures that the fish have been experiencing for months (I would hesitate to call them the acclimation temperature, since temperature fluctuated from day to day in accordance with ambient conditions), and not the temperature at which metabolic rates were measured (which was constant, and the same for all fish). It is well known that (testing) temperature affects SMR and MMR differently, but what you are showing is something more interesting and novel, so the word ‘temperature’ needs to be less ambiguous when used in conjunction with measurements of metabolic rate.

Response 1.3 – This is a good point. We now refer to *“thermal regime”* instead of *“temperature”* throughout when discussing effects of our temperature manipulation treatments to highlight that we are referring to the chronic differences rather than acute differences. We have carefully revised the manuscript where appropriate with this in mind to make it clear that we are focused on the long-term thermal environment. We have also revised the title of the manuscript to reflect this, with the revision now entitled: *“Associations between metabolic traits and growth rate in brown trout (Salmo trutta) depend on thermal regime”*.

Specific points:

Lines 6-9: The impression is given in this sentence that these relationships were only found in the warm treatment fish – the next sentences indicate that this was not the case, but readers may get confused.

Response 1.4 – We acknowledge that the above-mentioned sentences were unclear and have rephrased lines 6 – 9 in the revision to avoid any confusion.

Ln 6: “*We show that associations between SMR, MMR, and growth rate (a key fitness-related trait) vary depending on thermal regime (a potential selective agent) in offspring of wild-sampled brown trout from two populations reared for ~15 months in either a cool or warm (+ 1.8°C) regime.*”

Line 75: Reference 68 is relevant here, also (Robertsen et al. 2019).

Response 1.5 – We have added the suggested references.

Line 83: I feel we need something more targeted here by way of predictions – how might an increase in temperature be expected to alter the relationship between SMR or MMR and growth? At present the Introduction rather reads like ‘temperature affects metabolic rate in predictable ways; metabolic rate affects growth, but this depends on resource availability; therefore we investigated whether temperature influences how metabolic rate influences growth (but don’t say what level of resource availability we used, nor do we make any predictions as to which way any effect might go)’. This makes the introduction very bland and rather frustrating. The end of the penultimate paragraph emphasises the importance of resource availability in determining whether a given metabolic rate is beneficial or not – but this point is left hanging when you then go on to describe your own experiment, and you give no clue as to what you expected to find.

Response 1.6 – We apologise if our introduction was somewhat vague as to our specific motivations and expectations. We have revised the last two paragraphs of the Introduction to highlight why we believe our study is important, and to outline our hypotheses more explicitly. We hope this addresses the points you have raised, making for more engaging reading.

Ln 84: “*Research has focused largely on the role of resource availability as a selective agent, but other abiotic or biotic factors might also influence relationships between metabolic phenotypes and fitness components. Given the importance of temperature for energetic processes, thermal regime is potentially an important mediator of metabolic rate – fitness links, yet this has rarely been tested (30) despite the imperative of widespread climate warming.*

Here, we experimentally reared F1 offspring from two wild trout populations under different thermal regimes to explore how metabolic traits and their relationships with growth (a key fitness-associated trait) are mediated by chronic temperature increases. Specifically, we aimed to: (i) test whether long-term temperature elevation leads to variation in SMR and MMR; (ii) explore how metabolic traits are related to growth under different thermal regimes; and (iii) test whether the covariation of metabolic traits is influenced by thermal regime.

We expected that chronic temperature elevation would lead to compensatory responses in metabolism (lower SMR or higher MMR). We also expected that relationships between metabolism and growth might depend on thermal regime, whereby SMR and MMR would generally show a positive relationship to growth (under a natural thermal regime) as per the increased intake hypothesis (2), but that long-term temperature elevation might result in a negative SMR-growth relationship (if relatively high SMR are disadvantageous in warmer environments because less energy remains for growth once maintenance costs are paid, in line with the compensation hypothesis (2)). We expected that the MMR – growth relationship could either stay positive in the warmer regime, or alternatively could become neutral or negative (if SMR and MMR are tightly coupled, and lower SMR also results in reduced MMR).”

Line 102: Please indicate the number of broodstock females and males used from each source population.

Response 1.7 – We now include this detail at Ln 112.

Ln 112: *“Four males and three females were used as brood stock from Tawnyard, and twelve males and females from Srahrevagh.”*

Lines 114-115: The wording here is very ambiguous – the first part of the sentence states that 140 fish were put in one tank (presumably leaving 3 tanks empty), while the latter part says 35 fish were put in each of the four tanks. Which is correct?

Response 1.8 – We recognise this description was unclear and have corrected this in the revision.

Ln 127: *“... fry were allocated to four 203L capacity tanks in a larger experimental RAS (initial n = 35 per tank) ...”*

Line 116: I don’t understand the statement about rearing the populations separately ‘to prevent emergence of dominance hierarchies’. Trout form dominance hierarchies even if

they are all from the same population. Moreover, are you saying that fish from one population are dominant over the other? This requires explanation.

Response 1.9 – We simply meant to convey that while we had no expectations as to whether one population would be dominant over another, we reared populations separately for the duration of study to avoid any behavioural interactions among fish from different populations that might add an additional source of variation beyond our experimental manipulations. We recognise this was rather vague and confusing and have removed the reference to “*dominance hierarchies*” from Ln 129 of the revision.

Line 118: Clarify how the fish were fed (by hand or automatic feeder, one or more meals per day, food being spread or localised), since this has a major impact on the evenness of food distribution among fish.

Response 1.10 – We now state in the revision that fish were fed by automatic feeder above each tank, with a fixed quantity of feed released in pulses each day. We note that the presence of excess feed removed from tanks by filtration and during cleaning indicated that fish were feeding to satiation.

Ln 133: “*Automatic feeders above each tank delivered daily feed in multiple localised pulses. Excess feed removed during cleaning indicated fish were feeding to satiation.*”

Lines 126-127: I am presuming that this is a 2 x 2 design (2 populations x 2 temperatures), with one tank per treatment. Please clarify.

Response 1.11 – This is now clarified at Ln 140. We note that we considered “treatment” to refer to warm or cool thermal regime, which was replicated in two populations.

Ln 140: “*Each of the four tanks was allocated to one of two temperature treatments in January 2017, with one warm and one cool tank for each population*”

Lines 129-130: It is ambiguous as to whether the two catchments have the same seasonal temperature cycle (so that on a given day the two populations in the cool treatment were kept at the same temperature), or whether each population has its own tailored version of a cool and a warm temperature regime. I am guessing that the former is true (since the latter would be very complex to generate), but the wording actually implies the latter.

Response 1.12 - We have reworded Ln 141 to better convey that a single warm and cool temperature regime was created.

Ln 141: “Two thermal regimes were established by passing mixed water through one of two conditioning units. One created a cool treatment (both populations experienced the same seasonally-varying natural regime mimicking temperatures in the west of Ireland). The second unit created a warm treatment: $1.8\text{ }^{\circ}\text{C} \pm 0.55$ (SD) above the cool treatment.”

Lines 142-144: This is a bit confusing – growth rates were calculated ‘across the study period’, but the last measurement of size seems to have been in November 2017, which is 3 months before the measurement of metabolic rates and 5 months before the end of the study (according to line 112). I am presuming that the subset of fish subjected to respirometry must have been measured in February 2018 – maybe all were measured at the same time? Fig S3 also suggests that measurements continued until April 2018. Please clarify.

Response 1.13 – We apologise for this omission on our part – all fish were measured at the end of the experiment in April 2018, while just the subset that underwent respirometry were measured in February 2018. We now state this at Ln 156 and we have also included a new figure in the Supplementary Material (Figure S2) to provide more clarity on data collection and experimental design .

Ln 156: “... the fork length (mm) and mass (g) of lightly anaesthetised fish was recorded in April, June, July, September, and November (2017), and in April 2018 (when the study ended), with a subset of fish also measured in February 2018 during respirometry (see below) (Figure S2).”

Figure S2: Schematic of experimental design and timeline of data collection for the study. Offspring from two wild-origin populations were experimentally reared under two thermal regimes: a cool temperature treatment that mimicked natural temperatures; and a warm treatment, which was elevated by $1.8\text{ }^{\circ}\text{C}$ above the cool treatment.

Line 183: Note the typo – should be $P < 0.001$, not > 0.001 .

Response 1.14. – Corrected, thanks for picking up that error.

Line 199: Clarify which body mass this is – the mass at the time of measuring metabolic rate? Note that there is no mention of you having recorded body mass at any point in the experiment!

Response 1.15 – We have now clarified at Ln 186 that we included “*log₁₀body mass (at time of respirometry)*”. Our original submission stated that we also measured body mass when recording length (Ln 157 of the revision).

Line 195: ‘explored how temperature affected metabolic variation’ is not an ideal description of this analysis, since it could imply that you are referring to the temperature at which metabolic rate was measured (when you are actually referring to the rearing temperature, metabolic rates of all fish having been measured at the same temperature). Also it could imply that you are testing whether temperature affects the extent of variability in metabolic rates, when in fact you are testing if it affects the mean. Please rephrase.

Response 1.16 – Good point. We have rephrased at Ln 183 of the revision to make it clear we were testing the effects of thermal regime on the mean rather than acute temperature (*c.f.* Response 1.2).

Ln 183: “*We explored how thermal regime influenced mean values for metabolic traits (Aim 1) using two linear models (normal errors).*”

Line 214: The order in which the Results are presented seems slightly illogical to me. The analysis of growth rates is heavily focussed on the effect of metabolic rate, yet we are not told about what influences metabolic rate until later. It might be better to show the analyses of factors influencing metabolic rate first, and then go on to look at the consequences of this variation (i.e. for growth).

Response 1.17 – We have taken this point on board and have reordered our results in line with the suggestions. We have also rearranged the order of the methods to reflect this change and make for more coherent flow of the manuscript.

Lines 227-229: Fig S3 reveals something that could usefully be brought out: it would appear to be the case that warm treatment fish initially grew faster than cool (up until the summer), but then grew more slowly. This would make intuitive sense, since the warm treatment initially brought fish closer to the optimal temperature for growth in trout, but would then (by summer) have taken them further above it than the cool treatment. Adding temperature x time interactions to the model described here and in Table S4 would test for this. We need some exploration of this changing pattern, because readers will otherwise be puzzled why a temperature treatment seems to have no overall effect on growth (i.e. there is no main effect of temperature in the final model presented in Table 1). Furthermore, Fig. S3 also suggests that, regardless of thermal regime, low SMR fish had faster growth rates than high SMR fish at the start of the experiment (in the spring), but this gradually reversed so that by the end of the experiment (late winter) the growth rate was markedly higher in high SMR fish than in low. This striking pattern deserves some comment/discussion – is a low SMR beneficial at times when fish are growing little, but did the pattern reverse towards the following spring as small (i.e. high SMR) fish showed compensatory growth in the period leading up to smolting, as has been repeatedly shown in presmolt salmon (Nicieza & Braña 1993; Sigourney et al. 2013; Thomson & Lyndon 2018; Simmons et al. 2020)?

Response 1.18 - These are both interesting points, thank you for encouraging us to explore these aspects of our study in more detail. We now test for an interactive effect of temperature and time (Ln 198, Ln 220). The significant interaction term suggests that temperature effects were variable across the study, as you had suggested, and we have included some discussion of this finding at Ln 312.

As to the second point, we agree that this is also an interesting pattern, and have included some additional discussion on this at Ln 319 of the revision.

Ln 312: *“Temporal variation in the effects of thermal regime on growth suggested that the influence of temperature on growth is strongest when fish are closer to their thermal limits (~18 °C). During the summer months of our study, rearing temperatures in the warm treatment regularly exceeded the thermal growth optimum of ~13.9 °C for brown trout, and at times exceeded thermal growth limits, thus constraining growth. The negative SMR – growth association in the warm regime may have also served to limit growth during this period when fish in the warm treatment were pushed higher above their thermal optimum than those in the cool (66, 67).”*

Ln 319 *“Moreover, the benefits of a high metabolic rate for growth may have been limited in periods when growth potential is lower (e.g., in winter months when temperatures are sub-optimal and food availability reduced) but could have been most beneficial in the spring, when temperatures are closer to optimal ranges. Consideration of life-history might also be important here because metabolic rate tends to be positively related to migratory propensity/timing in salmonids, with individuals showing compensatory growth in the period prior to smolting (68). In general, metabolic phenotype, growth rates, and life-history traits likely co-evolve in relation to abiotic and biotic drivers that vary across space or through time (60, 69).”*

Line 324: It might be worth pointing out that there is perhaps no higher cost of a high MMR when temperatures are warm, because the animal is rarely operating at its MMR and is not forced to do so. This is quite different from the situation with SMR, which is an inevitable cost of living, and one that is higher at higher temperatures, leading to strong selection to reduce it.

Response 1.19 - Good point, we now mention this in the revision at Ln 373.

Ln 373: *“Additionally, since fish infrequently operate at MMR (77), the costs of maintaining a high MMR at warm temperatures are likely small relative to SMR, which is an unavoidable cost of living and thus potentially subject to stronger selection with warming.”*

Lines 332-342: You start this section saying that you are cautious about making population-level inferences – then go on to make them! I feel this is very speculative given that n=1 for migrant vs. resident populations, especially since the small pool of adults used to create the experimental populations. I suggest dropping this section.

Response 1.20 – We accept that we are limited in our ability to make population inferences, and we have removed the suggested section from the revision.

Lines 634-635: The acronyms ‘AB’ and ‘Non-AB’ do not feature in the Figure, so are not needed here.

Response 1.21 – That was a hang-over from an old version of the MS; we have corrected this now in the revision.

Supplementary information: Please expand on the description of how MMR was calculated – the rate of oxygen depletion would presumably take a few seconds to increase

once the fish had been placed in the respirometer (since it takes that time for water to circulate around the closed system), so did you exclude the first few readings, or only take the highest readings?

Response 1.22 – We have now included some additional detail on measuring MMR in our revised manuscript SI. MMR was calculated as the most linear decline in oxygen during the measurement period (i.e., the initial “plateau” as the water circulated through the probe was discarded, with only the steepest successive readings used in MMR calculations).

“MMR was calculated as the most linear decline in oxygen recorded during the measurement period estimated by rolling regression using the respR package (i.e., the initial plateau while the water circulated through the system after the fish entered the chamber was discarded).”

Table S1: The sample size (n=32) is ambiguous – does this refer to the number from each population, or in total? It would be better to give the sample size for each treatment group. Were these sizes representative of the fish in each treatment, or were they selected to maximise the size overlap between treatments? If the former, then it is odd that there is no size difference between fish reared in warm and cool treatments

Response 1.23 – We apologise for any ambiguity on our part, and we have revised this to provide the sample size for each temperature treatment broken down by population (“*n = 8 per each temperature treatment and population background combination*”). We assume that the absence of notable size differences was due to either compensatory growth exhibited in the spring prior to the smolting period, or alternatively (and possibly more likely), the warm treatment might not always have been beneficial for growth (e.g., in summer months) leading to sub-optimal growth and similar ultimate sizes. Fish were haphazardly selected for respirometry and were representative of the treatments (excluding any particularly large or small outlier fish that could not be accommodated in the respirometry chambers due to size constraints, approx. 3-4 per tank).

I hope that these comments are helpful

Neil Metcalfe (I am happy to be identified to the authors)

Response 1.24 – We really appreciate the detailed comments, which have certainly helped us to improve the manuscript.

- Elliott, J.M. & Elliott, J.A. (2010) Temperature requirements of Atlantic salmon *Salmo salar*, brown trout *Salmo trutta* and Arctic charr *Salvelinus alpinus*: predicting the effects of climate change. *Journal of Fish Biology*, 77, 1793-1817.**
- Elliott, J.M., Hurley, M.A. & Fryer, R.J. (1995) A new improved growth model for brown trout, *Salmo trutta*. *Functional Ecology*, 9, 290-298.**
- Jonsson, B. & Jonsson, N. (2009) A review of the likely effects of climate change on anadromous Atlantic salmon *Salmo salar* and brown trout *Salmo trutta*, with particular reference to water temperature and flow. *Journal of Fish Biology*, 75, 2381-2447.**
- Nicieza, A.G. & Braña, F. (1993) Compensatory growth and optimum size in one-year-old smolts of Atlantic salmon (*Salmo salar*). *Production of juvenile salmon, *Salmo salar*, in natural waters* (eds R.J. Gibson & R.E. Cutting), pp. 225-237.**
- Robertsen, G., Reid, D., Einum, S., Aronsen, T., Fleming, I.A., Sundt-Hansen, L.E., Karlsson, S., Kvingedal, E., Ugedal, O. & Hindar, K. (2019) Can variation in standard metabolic rate explain context-dependent performance of farmed Atlantic salmon offspring? *Ecology and Evolution*, 9, 212-222.**
- Sigourney, D.B., Letcher, B.H., Obedzinski, M. & Cunjak, R.A. (2013) Interactive effects of life history and season on size-dependent growth in juvenile Atlantic salmon. *Ecology of Freshwater Fish*, 22, 495-507.**
- Simmons, O.M., Britton, J.R., Gillingham, P.K. & Gregory, S.D. (2020) Influence of environmental and biological factors on the overwinter growth rate of Atlantic salmon *Salmo salar* parr in a UK chalk stream. *Ecology of Freshwater Fish*, 29, 665-678.**
- Thomson, M. & Lyndon, A.R. (2018) Comparing anadromous brown trout *Salmo trutta* in small, neighbouring catchments across contrasting landscapes: What is the role of environment in determining life-history characteristics? *Journal of Fish Biology*, 92, 593-606.**

***Note that all line numbers refer to the revised version of the manuscripts unless otherwise stated

Reviewer #2: Comments to the Author(s)

An initial disclaimer: I do not have the expertise in fish experiments, but the study seems well designed given the constraints of experimentally working on large animals. I also do not have experience with measurements of SMR and MMR on fish. I have therefore reviewed the paper given my experience of the effects of short- and long-term warming on biological rates and the statistical analyses. In this manuscript, the authors look at the response of brown trout to a long-term difference in temperature. This is a highly important question given the current climate change emergency that the world is undergoing. Specifically, there is a lot of evidence of marine life moving to new environments as temperatures change, but less evidence of the ability of fish to acclimate to warmer temperatures. Can fish acclimate to warmer temperatures? Here, the authors try to tackle this challenge by linking metabolism traits to growth rates (a fitness related trait). The authors run a long-term experiment (~1 year) where some fish were kept at conditions ~1.8 °C higher than others. Growth was tracked in ~100 individuals every few months and metabolic traits were measured at the end of the experiment. The main finding is that there is an interaction between the effect of temperature and metabolic traits on growth rate. More interestingly, the effect flips between the metabolic ceiling (maximum metabolic rate) and the metabolic floor (standard metabolic rate), such that the standard metabolic rate is flexible and acclimates to optimise growth at two different statistical regimes. I enjoyed reading the manuscript and found the results compelling. The statistical analyses are well thought out and appear robust. I am especially impressed by the presence of the data analysis code in an online repository for publication. I downloaded it and was able to reproduce the plots and analysis. I have a few major issues (see below) mainly about limitations in the dataset. However, given the robustness of the analysis I do not think these should limit its publication. I would like to congratulate them on a well written manuscript.

Response 2.1 – Thank you for the positive reception of our study and for your constructive suggestions on how to improve the manuscript. In our revision, we now: (1) directly discuss the limitations of our dataset (in particular using a small number of metabolic measurements at single timepoint to infer temporal growth trends); and (2) provide additional clarity on study design and analysis, including the addition of an schematic to give an overview of study

timeline and data collected. Please see our detailed responses below, and thank you encouraging us to include a more comprehensive discussion of the limitations of our study.

Major comments

My main issue is the lack of metabolic rate measurements through time, and as such there are only 32 measurements across the entire growth rate dataset (~150). Having metabolic rate measurements through the entire experiment alongside growth rates would have of course been optimal (and the authors acknowledge this in the last paragraph of the discussion). This is especially important as the authors acknowledge that growth rate and metabolic rates are size dependent, so having a single value at the end of the experiment to link to growth rates when they were at a different size is less than ideal. Given the variability in individual growth rate through time, how much individual variation in metabolic traits do the authors think were missed by not having these measurements?

Response 2.2 – We acknowledge that our approach makes the simplifying assumption of metabolic rate at a single timepoint being representative of an individual’s metabolism generally, and thus can be used to infer relationships with traits (such as growth) at other timepoints. While we agree that metabolic rate measurements spanning the study period would be desirable, measuring metabolic rate at a single point in time is reasonably common in studies of fish metabolism (Metcalf *et al.* 2016). We attempted to account for potential sources of variation in the growth – metabolic links in our analysis, e.g., accounting for size-based differences in both growth and metabolic traits. We fully accept that we may have missed other sources of temporal variation in the metabolic rate – growth relationship, which we now more explicitly acknowledge in our revision at Ln 329. Nonetheless, there is a growing body of evidence suggesting relative differences in both growth rates (Nater *et al.* 2018) and metabolic rates (Metcalf *et al.* 2016; Reemeyer and Rees 2020) in fish are consistent across timescales comparable to our study, and links between growth rates and metabolic correlates (boldness/aggression) have also been shown to be stable through time (Biro *et al.* 2014), giving us confidence that our findings are reasonably representative of more general patterns. As such, we feel that our study provides a good first step for generating additional interesting questions and motivating exploration of the temporal nature of context-dependent metabolic rate - fitness associations.

Ln 329: *“However, it is important to note that we measured metabolic traits at a single point in time and in a subset of fish, which may not fully reflect temporal trends or within-individual variation in metabolism that could alter metabolic – growth rate associations. While relative*

metabolic rates among individual salmonids tend to be stable through time (70), absolute metabolic rates can vary considerably depending on a suite of factors (9, 61). In the wild, the links between metabolic traits and growth (and their context-dependence) may be further modulated by additional factors e.g., food supply (71), which in turn show spatiotemporal variability (29, 35). A natural extension to our study would be to explore metabolic - growth relationships measured at multiple timepoints and under varying conditions to test whether such links are temporally stable.”

I got a bit lost in the timescale of the study. I think the study would benefit from a schematic showing when all the measurements were taken and the number of points for each measurement.

Response 2.3 – We have now included a schematic as suggested (Figure S2, see below), which gives an overview of the key time points of the study and the number of data points. For each temperature treatment, we measured growth in 21 – 26 individuals of each population, while metabolic rate measurements were taken for a subset of 8 individuals from each population for each temperature treatment (32 fish in total, 8×2 temperatures \times 2 populations). Note that in our original manuscript, we had presented the growth – metabolic rate analysis only for this subset of respirometry fish. We recognise that this is a limitation of our study and discuss this aspect more explicitly in our revision (*cf.* Response 2.2). We also now include an additional analysis of growth trends across all study fish (i.e., not just fish measured for metabolic rates), finding similar overall effects of temperature treatment, and thus increasing our confidence that our metabolic rate – growth results are likely to underpin the general trends observed in the study (see Response 2.4 below).

Figure S2

Figure S2: Schematic of experimental design and timeline of data collection for the study. Offspring from two wild-origin populations were experimentally reared under two thermal regimes: a cool temperature treatment that mimicked natural temperatures; and a warm treatment, which was elevated by 1.8 °C above the cool treatment.

I got quite confused about the number of data points in the study (See minor comment Line 137). Think you need to state again on Line 171 that this is the 32 individuals fish for which metabolic trials were taken. And possibly just be more up front about the limits of the study when talking about the analysis or in the discussion in regards to using metabolic traits from the end of the experiment to predict growth rates taken months earlier.

Response 2.4 – We apologise for any confusion generated by a lack of clarity on our part. We now more clearly state at Ln 204 that this analysis was on the subset of fish that were measured for metabolic rates, rather than all study fish. As noted in Response 2.3, we had actually also measured growth in the larger group of tracked fish (21 – 26 fish per temperature and population combination). In our revision, we now also include an analysis that examines temperature effects on growth across all study fish (*c.f.* Response 2.3) prior to presenting the analysis of metabolism and growth in respirometry subset. We hope this both provides more clarity on the data underlying each analysis and also helps to strengthen the link between effects of our temperature treatments on growth and the metabolic processes that might underpin these patterns.

Ln 191: “*We explored how temperature treatment and metabolic phenotype influenced specific growth rates across the study period (Aim 2) within a mixed effects modelling framework using the nlme package (55). We first built a mixed effects model (normal errors) to examine how thermal regime influenced individual-level growth rates across all study fish. The model included fixed effects of temperature treatment, population background, and time (continuous variable corresponding to months since start of experiment, fitted as a third order polynomial to account for non-linearity of growth through time). We included a temperature × time interaction to test whether thermal regime effects varied across the study. Individual identity was included as a random effect to account for multiple measurements of individuals. Since growth rate is size dependent (56), we included initial fork length as a covariate in the models. We accounted for temporal autocorrelation of growth rates by modelling an autoregressive error structure as a first order lag function of time.*

We next tested how SMR and MMR influenced growth rate patterns across temperature treatments in the subset of fish that underwent respirometry trials, using a similar modelling framework...”

Minor comments

Line 7: By metabolic traits the authors mean SMR and MMR I assume. Maybe metabolic traits can be set up in the first sentence alongside defining SMR and MMR?

Response 2.5 – We now define metabolic traits at Ln 4, and specifically refer to SMR and MMR at Ln 6 to avoid any confusion.

Ln 4: “*...relationships between metabolic traits (standard metabolic rate (SMR), maximum metabolic rate (MMR), and aerobic scope) and fitness across contexts are unresolved. We show that associations between SMR, MMR, and growth rate (a key fitness-related trait) vary depending on thermal regime...*”

Line 20-22: Nice and dramatic opening as is usual for metabolism-related papers. :)

Response 2.6 –Not *too* dramatic, we hope!

Line 22: “metabolic traits” need to be defined better here. Maybe just bringing Lines 27-34 up?

Response 2.7 – We have rearranged the order of our opening lines as suggested, to better define metabolic traits from the outset.

Ln 22: “*Metabolic traits – standard metabolic rate (SMR), maximum metabolic rate (MMR), and aerobic scope (AS) - can vary dramatically within species, but for reasons that remain obscure (3). The baseline energetic demands of ectotherms are defined by SMR, which represents the minimum energetic costs of maintaining tissues and homeostasis in an organism that is inactive, unstressed, and non-digestive (4) (termed basal metabolic rate (BMR) in endotherms within their thermoneutral zone i.e., requiring minimal changes in metabolic heat loss/gain). MMR in contrast, refers to the highest rate of aerobic metabolism (i.e. oxygen transport and ATP production) that can be achieved (5). AS – the difference between an organism’s SMR and MMR – determines the potential energy that can be allocated towards important functions including digestion, activity, growth and reproduction (6, 7).*”

Line 28: define thermoneutral zone

Response 2.8 – We now define thermoneutral zone as “*requiring minimal changes in metabolic heat loss/gain*”.

Line 50: temperature effects... on metabolism?

Response 2.9 – We have clarified that Ln 51 refers to “*temperature effects on metabolism*”.

Line 52: feel like there needs to be some qualifier or references to state “acute effects are reasonably well researched”

Response 2.10 – We have added references to Schulte *et al.* (2011) and Schulte (2015) to support this statement.

Line 72: RMR is a typo. Should be BMR I think.

Response 2.11 – Thanks for spotting this, corrected.

Line 120: Are Skretting Ltd the food company? Reference at the end of the sentence feels like you’re referencing their method?

Response 2.12 – Yes Skretting Ltd. were the feed manufacturers - We now make it clearer that we followed the manufacturer’s instructions to calculate daily pellet rations.

Ln 131: “*...calculated as a percentage of body mass as per manufacturer’s instructions...*”

Line 123-124: Really appreciate the transparency and honesty here.

Response 2.13 – Thanks!

Line 137: Seems to be some confusion of using populations in the study. For example Line 115 it says there are two tanks per population. Line137: population becomes each individual tank. Would suggest being really explicit about this early on

Response 2.14 – We apologise for any lack of clarity regarding our experimental design, we have now carefully rephrased Ln 127 and Ln 140. We hope our design is now made clearer to the reader (with the help of the supplementary schematic), whereby our main treatments of interest were the two different thermal regimes, each of which we replicated in two populations that were reared separately.

Ln 127: “*In December 2016, fry were allocated to four 203L capacity tanks in a larger experimental RAS (n = 35 per tank) with the populations reared separately throughout the study (i.e., each population allocated across two of the four tanks).*”

Ln 140: “*Each of the four tanks was allocated to one of two temperature treatments in January 2017, with one warm and one cool tank for each population (Figure S2).*”

Line 143: As well as dates it would be useful to know how long the study has been going on.

Response 2.15 - We have now included this detail in our schematic overview (Figure S2) in the revision.

Line 171-174: Just to be absolutely clear the raw values of SMR and MMR were regressed against body mass (all log₁₀). The values that went forward into the main analysis were the residuals of this analysis? Also were these residuals correlated (i.e. did individuals with higher residuals for MMR have higher residuals for SMR?). This information is present on Line 246 but it might be useful as a Supplementary plot.

Response 2.16 – Yes, we took the residuals from the log₁₀ transformed SMR/MMR and body mass relationships to give mass-independent estimates of metabolic traits. The relationship between rSMR and rMMR was positive but weak (see Response 2.18) and we have now included a supplementary figure in the revision to illustrate this (Figure S5, see below).

Figure S5: Residuals from the linear regressions used to given mass-independent estimates standard and maximum metabolic rate, where rSMR is the residuals from the relationship between \log_{10} standard metabolic rate and \log_{10} body mass and rMMR is the residuals from the \log_{10} maximum metabolic rate - \log_{10} body mass relationship.

Line 194: If you're always using a normal error structure, you could just say linear mixed effects models throughout, instead of GLMs.

Response 2.17 – We have now rephrased as suggested.

Lines 203 - 205: Do you have an a priori expectation for whether SMR or MMR should drive each other? If not, then Standardised Major Axis Regression might be used here. Granted though I have not looked at whether it can be used with covariates. smatr is a very well made R package to do this though.

Response 2.18 – Good suggestion, Standardised Major Axis Regression actually seems like a good fit here, as we considered it equally plausible that SMR could drive MMR (e.g., a lower

SMR limits MMR), or MMR could drive SMR (where a higher MMR requires higher SMR, e.g., due to requirements for larger organ sizes etc.) We now use Standardised Major Axis Regression in our revised manuscript, finding the same qualitative results (no strong coupling of SMR and MMR with no temperature effect).

Ln 223: “*Lastly, we tested whether temperature treatment influenced the relationships between metabolic traits (Aim 3). We used standardised major axis regression for this analysis using the smatr package (60) because we had no a priori expectations as to which metabolic trait should drive the other (i.e., rather than predicting MMR from SMR or vice versa, we assumed the relationship could be symmetric, where either variable could be on either axis).*”

Ln 272; “*There was a weak, non-significant coupling of rMMR and rSMR in both thermal regimes ($P = 0.068$ and $P = 0.100$ in Cool and Warm respectively), with no effect of thermal regime on the slope ($\chi^2 = 2.00$, $df = 1$, $P = 0.157$) or intercept of the relationship ($\chi^2 = 0.834$, $P = 0.361$) (Figure S4).*”

Line 209: Would use $p < 0.05$ than %

Response 2.19 – Corrected.

Figure 1: in (a) could you highlight the key results here? The rSMR * Warm and rMMR*warm results. Shade that background a tiny bit or put stars by the factors? Would allow readers to more easily find the key result

Response 2.20 – We have placed a star next our two key results in this Figure in our revised manuscript (note that Figure 1 now appears as Figure 2 in our revision based on a comment from Reviewer #1 suggesting we re-order the results).

Figure 1: b and c. I always struggle when there are predictions and no points on the plot when it is not a theoretical model. Could you possibly put on partial residuals? These can be taken from a visreg object - <https://pbreheny.github.io/visreg/> - and are used to show the impact of a predictor given other covariates. This is very similar to how the predictions were created. If nothing else it will allow the reader to see how many points underpin the dataset.

Response 2.21 – Sure (and thanks for the suggested resource). We have now added partial residuals to Figure 2 panels B and C in our revision (see below). Note that we also tested for the sensitivity of our results to an outlier point that we identified as having a high residual growth rate value (> 0.3 % day⁻¹) by re-running our analysis without this point. Our results

were qualitatively similar to our original analysis (see additional Figures below), so we retained the outlier point in the final analysis we present in the manuscript.

Figure 2

Figure 2: (A) Coefficient estimates (\pm 95% confidence intervals) from the mixed effect models describing the effects of residual standard metabolic rate (rSMR), maximum metabolic rate (rMMR), aerobic scope (rAS), and temperature treatment (Cool and Warm) on specific growth rates of brown trout from two populations (anadromous, and non-anadromous) across the study (“Time” = months since initiating treatments). Interactive effects of thermal regime on metabolic rates are highlighted by red stars. (B) Predicted growth rates in response to marginal effects of rSMR and (C) rMMR, at each thermal regime (shaded regions show the 95% confidence intervals for the predictions). Growth rates were predicted at mean values for the remaining explanatory variables.

Sensitivity of our analysis to an outlier point identified in Figure 2:

The same trends were observed and the same interactions and main effects were retained in the model when the analysis was run with and without the outlier growth rate point (as identified in Figure 2 above). Below are model predictions for (A) SMR and (B) MMR based on the dataset with the outlier point excluded.

References

- Biro PA, Adriaenssens B, and Sampson P. 2014. Individual and sex-specific differences in intrinsic growth rate covary with consistent individual differences in behaviour. *J Anim Ecol* **83**: 1186–95.
- Metcalfe NB, Van Leeuwen TE, and Killen SS. 2016. Does individual variation in metabolic phenotype predict fish behaviour and performance? *J Fish Biol* **88**: 298–321.
- Nater CR, Rustadbakken A, Ergon T, *et al.* 2018. Individual heterogeneity and early life conditions shape growth in a freshwater top predator. *Ecology* **99**: 1011–7.
- Reemeyer JE and Rees BB. 2020. Plasticity, repeatability and phenotypic correlations of aerobic metabolic traits in a small estuarine fish. *J Exp Biol* **223**.
- Schulte PM. 2015. The effects of temperature on aerobic metabolism: Towards a mechanistic understanding of the responses of ectotherms to a changing environment. *J Exp Biol* **218**: 1856–66.
- Schulte PM, Healy TM, and Fanguie NA. 2011. Thermal performance curves, phenotypic plasticity, and the time scales of temperature exposure. *Integr Comp Biol* **51**: 691–702.

Appendix C

University College Cork
Distillery Fields Campus
North Mall
Cork
Ireland

17th August 2021

Dear Prof. Carvalho,

Many thanks for your letter dated 16th August 2021, offering acceptance pending minor revisions of our resubmitted manuscript (RSPB-2021-1509 “*Associations between metabolic traits and growth rate in brown trout (Salmo trutta) depend on thermal regime*”). Below, we have addressed the outstanding reviewer comments, with reference to the changes we have made in the revised text. We are also attaching two Word documents as part of this resubmission, one entitled “Archer et al Metabolic traits and growth Accepted with Minor Revision (Track Changes)”, and a second version with all changes accepted (“Archer et al Metabolic traits and growth Accepted with Minor Revision), along with the supporting information for the article. We hope this version of the manuscript will now be suitable for publication in *Proceedings of the Royal Society B*.

None of the material contained in the revised manuscript has been published or is under consideration elsewhere, including the internet. Thank you for your consideration and please do not hesitate to contact me if you have any questions.

Yours faithfully,

Louise Archer
University College Cork
E-mail: l.archer@umail.ucc.ie

*** Note that all line numbers refer to the revised version of the manuscript unless otherwise stated

Reviewer #1 (Comments to the Authors):

The authors have done an excellent job in responding to the points raised by myself and the other reviewer, and as a result the manuscript is much improved. I only have a small number of points, mostly relating to typos:

Line 79: Should be ‘may be’ not ‘maybe’.

Response 1.1 – Corrected, thanks for spotting this.

Line 209-210: Slight re-phrasing to aid clarity: ‘...rSMR and temperature, and between rMMR and temperature, to test if...’

Response 1.2 – We have made the suggested change.

Line 211: Insert ‘the’ – ‘..included the two-way interaction...’

Response 1.3 – Corrected in the revision.

Lines 312-318: The factual statements about trout thermal biology need to be more clearly referenced – at present it is not clear where you get the information on the thermal growth optimum nor the thermal growth limits for trout. Indeed, the two Elliott references (refs 66 and 67) are quoted as if they referred to your experiment. Also you need to clarify that the ‘thermal limit’ quoted on lines 312-313 is that for growth (trout can temporarily survive much higher temperatures than 18C).

Response 1.4 – We have added a citation to Elliott and Elliott (2010) at Ln 314 of the revision (ref. # 66) to support our statements on the thermal growth limits of trout. We have also moved citations to Elliott *et al.* (1996) and Elliott and Hurley (2000) (refs. 67, 68) to Ln 316 and Ln 323 to make it clearer that these references relate to previously documented thermal growth optima of trout.

At Ln 314 of the revision, we also now clarify that we refer to “*thermal growth limits...*” rather than critical limits for survival.

Line 319: ‘have been’ not ‘have be’.

Response 1.5 – Corrected, thanks for spotting this.

Line 329: Delete ‘and in a subset of fish’ – the sample size is not relevant to the arguments you are making here, which relate only to the single measurement per individual.

Response 1.6 – Good point. We have now made the suggested change in the revision.

References

Elliott JM, Elliott JA. Temperature requirements of Atlantic salmon *Salmo salar*, brown trout *Salmo trutta* and Arctic charr *Salvelinus alpinus*: Predicting the effects of climate change. *Journal of Fish Biology*. 2010;77(8):1793–817.

Elliott JM, Hurley MA, Fryer RJ. A new, improved growth model for brown trout, *Salmo trutta*. *Functional Ecology*. 1995;9(2):290–8.

Elliott JM, Hurley MA. Daily energy intake and growth of piscivorous brown trout, *Salmo trutta*. *Freshwater Biology*. 2000;44(2):237–45.